# Two Sides of the Same Coin: Heterophily and Oversmoothing in Graph Convolutional Neural Networks

## Abstract

In node classification tasks, heterophily and oversmoothing are two problems that can hurt the performance of graph convolutional neural networks (GCNs). The heterophily problem refers to the model's inability to handle heterophilous graphs where neighboring nodes belong to different classes; the oversmoothing problem refers to the model's degenerated performance with increasing number of layers. These two seemingly unrelated problems have been studied mostly independently, but there is recent empirical evidence that solving one problem may benefit the other. In this work, beyond empirical observations, we aim to: (1) analyze the heterophily and oversmoothing problems from a *unified* theoretical perspective, (2) identify the *common causes* of the two problems based on our theories, and (3) propose simple yet effective *strategies* to address the common causes.

In our theoretical analysis, we show that the common causes of the heterophily and oversmoothing problems—namely, the relative degree of a node (compared to its neighbors) and its heterophily level—trigger the node representations in consecutive layers to "move" closer to the original decision boundary, which increases the misclassification rate of node labels under certain constraints. We theoretically show that: (1) Nodes with high heterophily have a higher misclassification rate. (2) Even with low heterophily, degree disparity in a node's neighborhood can influence the movements of node representations and result in a "pseudo-heterophily" situation, which helps to explain oversmoothing. (3) Allowing not only positive, but also negative messages during message passing can help counteract the common causes of the two problems. Based on our theoretical insights, we propose simple modifications to the GCN architecture (i.e., learned degree corrections and signed messages), and we show that they alleviate the heteorophily and oversmoothing problems with extensive experiments on nine real networks. Compared to other approaches, which tend to work well in either heterophily or oversmoothing, our modified GCN model performs well in both problems.

## 1 Introduction

In recent years, GCNs (Defferrard et al., 2016; Kipf & Welling, 2016; Veličković et al., 2017) have been widely used in applications ranging from social science (Li & Goldwasser, 2019) and biology (Yan et al., 2019) to program understanding (Allamanis et al., 2018; Shi et al., 2019). A typical GCN architecture (Gilmer et al., 2017) for the node classification task can be decomposed into two main components: propagation/aggregation, and combination. Messages are first exchanged between neighboring nodes, then aggregated. Afterwards, the messages are combined with the self-representations (a.k.a., the current node representations) to update the node representations. Though GCNs are effective, they have some key limitations.

The first limitation is the "oversmoothing" problem (Li et al., 2018): the performance of GCNs degrade when stacking many layers. Recent work has found that oversmoothing could be caused by GCNs exponentially losing expressive power in the node classification task (Oono & Suzuki, 2019) and that the node representations converge to a stationary state which is decided by the degree of the nodes and the input features (Chen et al., 2020; Wang et al., 2019; Rong et al., 2019; Rossi et al., 2020). These works focus on the analysis of the steady state in the limit of infinite layers, but they do not explore the dynamics on how oversmoothing is triggered, or which nodes tend to cause it. Other works like (Xu et al., 2018; Wang et al., 2019; Chen et al., 2020) focus on heuristic-based model designs to alleviate the oversmoothing problem. To fill this gap, we seek to theoretically analyze the dynamics around oversmoothing. In our empirical analysis, by dividing the nodes in real graphs into

several degree groups and tracking their average accuracy while increasing the layers of GCN (Kipf & Welling, 2016), we observe two phases (Fig. 1): the initial stage when higher-degree nodes have higher accuracy and the developing stage when the accuracy of higher-degree nodes decreases more sharply (cf. § 5.4 for details). As we will show theoretically and empirically, the low-degree nodes are the ones that trigger oversmoothing. Though Chen et al. (2020) provide some explanations for the developing stage by analyzing the convergence rate, their theory fails to characterize the initial stage. In fact, we find that the analysis of initial stage provides valuable insights about oversmoothing.

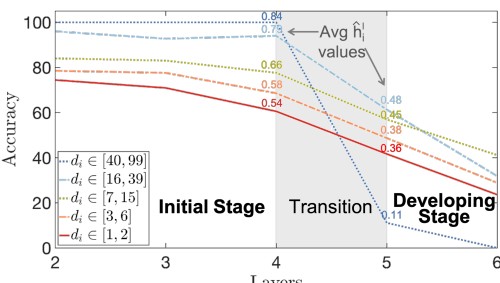

The second limitation of GCNs is their poor performance on heterophilous graphs (Pei et al., 2019; Lim et al., 2021), which—unlike homophilous graphs—have many neighboring nodes that belong to different classes (Newman, 2002). For instance, in protein networks, amino acids of different types tend to form links (Zhu et al., 2020), and in transaction networks, fraudsters are more likely to connect to accomplices than to other fraudsters (Pandit et al., 2007). Most GCNs (Kipf & Welling, 2016; Veličković et al., 2017) fail to effectively capture heterophily, so emerging works have proposed some effective designs (Pei et al., 2019; Zhu et al., 2020), including signed messages (positive and negative interactions between nodes) proposed by concurrent works (Chien et al., 2021; Bo et al., 2021).

Figure 1: Accuracy of nodes grouped by degree $d_i$ on Citeseer. Initial stage: when mean effective homophily $\widehat{h}_i^l$ (ratio of a node's neighbors in the same class–§ 3.3.2) is high, the accuracy increases as the degree increases. Developing stage: when $\widehat{h}_i^l$ is low, the accuracy of high-degree nodes drops more sharply.

Though these works justify signed messages from a spectral perspective, they focus on asymptotic results, requiring an infinite number of filters. Our work does not make this assumption, and our theory works in more practical scenarios (§ 3.4).

These two limitations have mostly been studied independently, but recent work on oversmoothing (Chen et al., 2020) was shown empirically to address heterophily, and vice versa (Chien et al., 2021). Motivated by this empirical observation, we set out to understand the theoretical connection between the oversmoothing and heterophily problems by studying the change in node representations during message passing. Specifically, we make the following contributions:

- **Theory:** We introduce the first theory that explains the connections between heterophily and oversmoothing which opens up the possibility of jointly studying the two problems.
- **Insights:** We find that under conditions, the relative degree of a node (compared to its neighbors) and its heterophily level affect the misclassification rate of node labels in the oversmoothing and heterophily problems. We further prove that signed messages can mitigate their negative impact.
- **Improved model & empirical analysis:** Based on these insights, we design a generalized GCN model, GGCN, that allows negative interactions between nodes and compensates for the effect of low-degree nodes through a learned rescaling scheme. Our empirical results show that our model is robust to oversmoothing, achieves state-of-the-art performance on datasets with high levels of heterophily, and achieves competitive performance on homophilous datasets.

## 2 PRELIMINARIES

We first provide the notations & definitions that we use in the paper, and a brief background on GCNs.

**Notation.** We denote an unweighted and self-loop-free graph as $\mathcal{G}(\mathcal{V}, \mathcal{E})$ and its adjacency matrix as $\mathbf{A}$. We represent the degree of node $v_i \in \mathcal{V}$ by $d_i$, and the degree matrix—which is a diagonal matrix whose elements are node degrees—by $\mathbf{D}$. Let $\mathcal{N}_i$ be the set of nodes directly connected to $v_i$, i.e., its neighbors. $\mathbf{I}$ is the identity matrix. We denote the node representations at $l$-th layer as $\mathbf{F}^{(l)}$, and the $i$-th row of $\mathbf{F}$ is $\mathbf{f}_i^{(l)}$, which is the representation of node $v_i$. The input node features are given by $\mathbf{F}^{(0)}$. The weight matrix and bias vector at the $l$-th layer are denoted as $\mathbf{W}^{(l)}$ and $\mathbf{b}^{(l)}$, respectively.

**Homophily and Heterophily.** Given a set of node labels/classes, *homophily* captures the tendency of a node to have the same class as its neighbors. Specifically, the homophily of node $v_i$ is defined as $h_i \equiv \mathbb{E}\left(\frac{|\mathcal{N}_i^s|}{|\mathcal{N}_i|}\right)$, where $\mathcal{N}_i^s$ is the set of neighboring nodes with the same label as $v_i$, $|\cdot|$ is the cardinality operator, and the expectation is taken over the randomness of the node labels. High homophily corresponds to low heterophily, and vice versa, so we use these terms interchangeably.

**Supervised Node Classification Task.** We focus on node classification: given a subset of labeled nodes (from a label set $\mathcal{L}$), the goal is to learn a mapping $\mathscr{F} : \mathbf{f}_i^{(0)} \mapsto y_i$ between each node $v_i$ and its ground truth label $y_i \in \mathcal{L}$. We use $\hat{y}_i$ to represent the predicted label of $v_i$ and define the **misclassification rate** of a fixed learnt model over a node set $\mathcal{V}$ as: $\mathbb{E}\left(\frac{|\{v_i|\hat{y}_i \neq y_i, v_i \in \mathcal{V}\}|}{|\mathcal{V}|}\right)$. The expectation is taken over the randomness of graph structures and input node features.

**Graph Convolutional Neural Networks.** In node classification tasks, an $L$-layer GCN contains two components (Gilmer et al., 2017): (1) neighborhood propagation and aggregation: $\widehat{\mathbf{f}_i^{(l)}} = \text{AGGREGATE}(\mathbf{f}_j^{(l)}, v_j \in \mathcal{N}_i)$, and (2) combination: $\mathbf{f}_i^{(l+1)} = \text{COMBINE}(\widehat{\mathbf{f}_i^{(l)}}, \mathbf{f}_i^{(l)})$, where AGGREGATE and COMBINE are learnable functions. The loss is given by $\mathscr{L}_{CE}=\text{CrossEntropy}(\text{Softmax}(\mathbf{f}_i^{(L)}\mathbf{W}^{(L)} + \mathbf{b}^{(L)}), y_i)$. The vanilla GCN suggests a renormalization trick on the adjacency $\mathbf{A}$ to prevent gradient explosion (Kipf & Welling, 2016). The $(l+1)$-th output is given by: $\mathbf{F}^{(l+1)} = \sigma(\tilde{\mathbf{A}}\mathbf{F}^{(l)}\mathbf{W}^{(l)})$, where $\tilde{\mathbf{A}} = \tilde{\mathbf{D}}^{-1/2}(\mathbf{I} + \mathbf{A})\tilde{\mathbf{D}}^{-1/2}$, $\tilde{\mathbf{D}}$ is the degree matrix of $\mathbf{I} + \mathbf{A}$, and $\sigma$ is ReLU. When the non-linearities in the vanilla GCN are removed, it reduces to a linear model called SGC (Wu et al., 2019), which has competitive performance and is widely used in theoretical analyses (Oono & Suzuki, 2019; Chen et al., 2020). For SGC, the $l$-th layer representations are given by: $\mathbf{F}^{(l)} = \tilde{\mathbf{A}}^l\mathbf{F}^{(0)}$ and the last layer is a logistic-regression layer: $\hat{y}_i = \text{Softmax}(\mathbf{F}^{(L)}\mathbf{W}^{(L)} + \mathbf{b}^{(L)})$. We note that only one weight matrix $\mathbf{W}^{(L)}$ is learned; it is equivalent to the products of all weight matrices in a linear GCN. Find more related works in § 6.

## 3 THEORETICAL ANALYSIS

In this section, we formalize the connection between the oversmoothing and heterophily problems in two steps: (1) We show how the movements of node representations relate to the misclassification rate of node labels; and (2) we identify the factors that affect the movements of representations. We use the SGC model to analyze binary node classification with nodes in class 1 denoted as set $\mathcal{V}_1$ and nodes in class 2 denoted as set $\mathcal{V}_2$. We show empirically (§ 5) that the insights obtained in this section are effective for other non-linear models in multi-class classification.

### 3.1 ASSUMPTIONS

Next, we use "i.d." to represent random variables/vectors that follow the same marginal distribution and their joint probability density function (PDF) is a permutation-invariant function $p(\mathbf{x}_1, \ldots, \mathbf{x}_n) = p(\mathbf{P}(\mathbf{x}_1, \ldots, \mathbf{x}_n))$, where $\mathbf{P}(\cdot)$ means permutation.

**(1) Random graph**: Node degrees $\{d_i\}$ are i.d. random variables. $\{(\cdot)_i\}$ represents a set with $i = 1, \ldots, |\mathcal{V}|$. **(2) Inputs**: (2.1) Node labels $\{y_i\}$ are i.d. Bernoulli random variables given by the ratio $\rho$: $\rho \equiv \frac{\mathbb{P}(y_i=1)}{\mathbb{P}(y_i=2)}, \forall i$. (2.2) Initial input node features $\{\mathbf{f}_i^{(0)}\}$ are i.d. random vectors given by (PDF) $f(\mathbf{x})$, which is expressed as: $f(\mathbf{x}) = \begin{cases} f_1(\mathbf{x}), & \text{when } y_i = 1. \\ f_2(\mathbf{x}), & \text{when } y_i = 2. \end{cases}$ $\mathbb{E}(\mathbf{f}_i^{(0)}|y_i) = \begin{cases} \boldsymbol{\mu}, & \text{when } y_i = 1. \\ -\rho\boldsymbol{\mu}, & \text{when } y_i = 2. \end{cases}$ so $\mathbb{E}(\mathbf{f}_i^{(0)}) = \mathbf{0}$. **(3) Independence**: $\{d_i\}$ are independent of $\{y_i\}$ and $\{\mathbf{f}_i^{(0)}\}$.

### 3.2 MISCLASSIFICATION RATE & MOVEMENTS OF NODE REPRESENTATIONS

The following lemmas illustrate why the movements of node representations are a good indicator of SGC's performance. We view the changes of node representations across the layers as their movements; i.e., $\mathbf{f}_i^{(l+1)}$ is moved from $\mathbf{f}_i^{(l)}, \forall i$. The misclassification rate of an $L$-layer SGC is determined by $\{\mathbf{f}_i^{(L)}\}$, which is the input to the last regression layer, and can be studied through the decision boundary of $\mathbf{f}_i^{(L)}$.

**Lemma 1** In SGC, the decision boundaries w.r.t. $\{\mathbf{f}_i^{(L)}\}$ in multi-class classification are linear hyperplanes. In particular, the decision boundary is a single hyperplane for binary classification.

*Proof.* We provide the proof in App. A.1. □

The decision boundary of an $(L+1)$-layer SGC can be viewed as a perturbation to the decision boundary of an $L$-layer SGC due to the movements from $\{\mathbf{f}_i^{(L)}\}$ to $\{\mathbf{f}_i^{(L+1)}\}$. Due to Lemma 1, we can use the hyperplane $\mathbf{x}\mathbf{w} + b = 0$ to represent the decision boundary of $\{\mathbf{f}_i^{(L)}\}$, where $\mathbf{w}$ is a unit vector, $b$ is a scalar, and the $\{\mathbf{f}_i^{(L)}\}$ are classified to class 1 if $\mathbf{f}_i^{(L)}\mathbf{w} + b > 0$. Suppose when $\mathbf{w} = \mathbf{w}^*$ and $b = b^*$, we find an optimal hyperplane $\mathbf{x}\mathbf{w}^* + b^* = 0$ that achieves the lowest

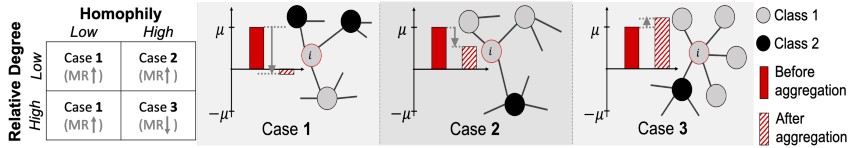

Figure 2: Node representation dynamics during neighborhood aggregation (in 1D for illustration purposes; 'MR': misclassification rate). The expectation of node representations from class 1 & 2 are denoted by $\mu$ and $-\mu$, respectively. The bars show the expected node representations of node $v_i$ before and after the aggregation.

total misclassification rate (§ 2) for $\{\mathbf{f}_i^{(L)}\}$. Due to the movements of node representations, the new optimal hyperplane is decided by: $\mathbf{x}\mathbf{w}^{*\prime} + b^{*\prime} = 0$. In Lemma 2, we show that, under constraints, moving towards the original decision boundary with a non-zero step results in a non-decreasing total misclassification rate under the new decision boundary.

**Lemma 2** Moving representations $\{\mathbf{f}_i^{(L)}\}$ from class 1 by adding $-t\mathbf{w}^{*T}$, $t > 0$, s.t. $\mathbf{w}^{*T}\mathbf{w}^{*\prime} > 0$, the new total misclassification rate is no less than the misclassification rate before the movements.

*Proof.* We provide the proof in App. A.2. □

Note that the special case where the representations of the two classes swap positions (e.g, bipartite graphs) violates the condition $\mathbf{w}^{*T}\mathbf{w}^{*\prime} > 0$ and leads to different conclusions. We refer to the condition $\mathbf{w}^{*T}\mathbf{w}^{*\prime} > 0$ as "non-swapping condition" and throughout the paper, we analyze SGC under that. Lemma 2 shows that, under the "non-swapping condition", if $\{\mathbf{f}_i^{(l)}\}$ move towards the original decision boundary (or the other class), SGC tends to perform worse.

### 3.3 MOVEMENTS OF NODE REPRESENTATIONS IN SHALLOW AND DEEPER LAYERS

In the following theorems, we will show how relative degree and homophily level will affect the movement of node representations in both shallow and deeper layers. We show results given the degree, $d_i$, of node $v_i$ and assume that $v_i$ is in class 1. The results in class 2 can be derived similarly.

#### 3.3.1 INITIAL STAGE: SHALLOW LAYERS

**Theorem 1.** *[Initial Stage] Under the "non-swapping condition", during message passing in shallow layers, the nodes with **low** homophily $h_i \leq \frac{\rho}{1+\rho}$ and the nodes with high homophily $h_i > \frac{\rho}{1+\rho}$ but **small** relative degree are prone to be misclassified. Defining $r_{ij} \equiv \sqrt{\frac{d_i+1}{d_j+1}}$ and the relative degree $\overline{r_i} \equiv \mathbb{E}_{\mathbf{A}|d_i, v_i \in \mathcal{V}_1}(\frac{1}{d_i}\sum_{j \in \mathcal{N}_i} r_{ij})$, the conditional expectation of representation $\mathbf{f}_i^{(1)}$ is given by:*

$$\mathbb{E}_{\mathbf{A},\{y_i\},\{\mathbf{f}_i^{(0)}\}|d_i, v_i \in \mathcal{V}_1}(\mathbf{f}_i^{(1)}|v_i \in \mathcal{V}_1, d_i) = \left(\frac{((1+\rho)h_i - \rho)d_i\overline{r_i} + 1}{d_i + 1}\right)\mathbb{E}(\mathbf{f}_i^{(0)}) \equiv \gamma_i^1 \mathbb{E}(\mathbf{f}_i^{(0)}), \quad (1)$$

*where the multiplicative factor $\gamma_i^1$ is:*

$$\gamma_i^1 \in \begin{cases} (-\infty, \frac{1}{2}], & \text{if } h_i \leq \frac{\rho}{1+\rho} \\ (0, 1], & \text{if } h_i > \frac{\rho}{1+\rho} \ \& \ \overline{r_i} \leq \frac{1}{(1+\rho)h_i - \rho} \\ (1, \infty), & \text{otherwise.} \end{cases} \quad (2)$$

*Proof.* We provide the proof in App. A.3. □

From Thm. 1, we identify three types of movements of node representations, which are characterized by relative degree $\overline{r_i}$ and homophily level $h_i$. For illustration purposes, in Fig. 2, we illustrate the three cases when we apply our theorem to 1D node representations. The bars reflect the value change of $v_i$'s node representation. Intuitively, under heterophily (case 1) or under high homophily but low degrees (case 2), the expected representations will move heavily towards the opposing class. Otherwise, they will move away from the opposing class (case 3). Among the three cases, only nodes with high homophily and high relative degree benefit from propagation and aggregation.

**Explanation of the heterophily and oversmoothing problems:**
- In heterophilous graphs, node representations move heavily towards the opposite class. According to Lemma 2, under the "non-swapping" condition, it indicates higher misclassification rate (i.e., poorer performance) for SGC.
- In homophilous graphs, high-degree nodes may initially benefit (case 3), leading to performance gain in shallow layers. However, the representations of low-degree nodes tend to move towards the opposing class (case 2) and eventually flip the labels. The misclassification of low-degree nodes causes a *pseudo-heterophily* situation (case 1) later.

### 3.3.2 DEVELOPING STAGE: DEEPER LAYERS

The following theorem analyzes the pseudo-heterophily scenario in deeper layers. Based on Thm. 1, message passing will result in scaling the node representations ($\gamma_i^1$) and some nodes may cross the origin. To account for those, let $\xi_i^l$ be an accumulated discount factor at the $l$-th layer and $\mathbb{E}_{\mathbf{A}, \{y_i\}, \{\mathbf{f}_i^{(0)}\}|d_i}\left(\frac{\mathbf{f}_i^{(l)}}{d_i+1}|d_i\right) = \xi_i^l\boldsymbol{\mu}$. Conditioned on $\mathbf{A}$ and $\xi_i^l$, the nodes contributing positively are defined as: $\hat{\mathcal{N}}_i^s(\mathbf{A}, \xi_i^l) \equiv \{v_j|v_j \in \mathcal{N}_i \text{ and } (\xi_i^l\boldsymbol{\mu}) \cdot \mathbb{E}_{\{y_i\}, \{\mathbf{f}_i^{(0)}\}|\xi_i^l, \mathbf{A}}\left(\mathbf{f}_j^{(l)^T}|\xi_i^l, \mathbf{A}\right) > 0\}$. We denote: $\mathbb{E}_{\mathbf{A}, \{y_i\}, \{\mathbf{f}_i^{(0)}\}|d_i, \xi_i^l, v_j \in \mathcal{N}_i}\left(\mathbf{f}_j^{(l)}r_{ij}|d_i, \xi_i^l, v_j \in \mathcal{N}_i\right) = \begin{cases} \xi_i^{l'}\boldsymbol{\mu} & v_j \in \hat{\mathcal{N}}_i^s(\mathbf{A}, \xi_i^l) \\ -\rho_i^l\xi_i^{l'}\boldsymbol{\mu} & v_j \notin \hat{\mathcal{N}}_i^s(\mathbf{A}, \xi_i^l) \end{cases} \cdot \xi_i^{l'}$ and $\rho_i^l$ characterize the neighborhood property of $v_i$. We further define the **effective homophily** of node $i$ as $\hat{h}_i^l = \mathbb{P}(v_j \in \hat{\mathcal{N}}_i^s|v_j \in \mathcal{N}_i, d_i, \xi_i^l)$. Intiuitively, even if the original graph is not heterophilous, from the perspective of message passing under this pseudo-heterophily scenario, the deeper layers will act as if the graph is more heterophilous than it truly is, which may increase the misclassification rate.

**Theorem 2.** *[Developing Stage] Under the "non-swapping condition", during message passing in deeper layers, when the effective homophily $\hat{h}_i^l$ is sufficiently **low** ($\hat{h}_i^l \leq \frac{\rho_i^l}{1+\rho_i^l}$), the nodes with **higher** effective relative degree are prone to misclassification. Specifically, by defining the effective relative degree as $\bar{r}_i^l \equiv \frac{\xi_i^{l'}}{\xi_i^l}$, the conditional expectation of $\mathbf{f}_i^{(l+1)}$ is:*

$$\mathbb{E}_{\mathbf{A}, \{y_i\}, \{\mathbf{f}_i^{(0)}\}|d_i, \xi_i^l}\left(\mathbf{f}_i^{(l+1)}|d_i, \xi_i^l\right) = \frac{\left((\hat{h}_i^l(1+\rho_i^l) - \rho_i^l)d_i\bar{r}_i^l + 1\right)}{d_i + 1}\xi_i^l\boldsymbol{\mu} \equiv \gamma_i^{(l+1)}\mathbb{E}(\mathbf{f}_i^{(0)}). \quad (3)$$

*Proof.* We provide the proof in App. A.4. Other nodes can be derived similarly. $\square$

We note that Eq. (3) holds for layers $l \geq 2$ in general. We regard layers for which $\hat{h}_i^l > \frac{\rho_i^l}{1+\rho_i^l}$ holds for most nodes $i$ as part of the initial stage, as they exhibit three cases similar to Eq. (2).

### 3.4 SIGNED MESSAGES & MOVEMENTS OF REPRESENTATIONS

In this section, we provide theory to show that signed messages can help enhance the performance in heterophilous graphs and also alleviate oversmoothing. Due to limited space, we only show the effect of signed messages in the initial stage; similar results can be derived in the developing stage.

**Setup.** Signed messages consist of the negated messages sent by neighbors of the opposing class (i.e., after multiplying initial messages by -1), and the unchanged messages ("positive messages") sent by neighbors of the same class. In reality, ground-truth labels are not accessible for every node, so we use approximations, which introduces errors. For node $v_i$, we define $m_i^l$ as the *ratio of neighbors that send incorrect messages* at the $l$-th layer (i.e., different-class neighbors that send non-negated messages and same-class neighbors that send negated messages). We define the $l$-th layer *error rate* as $e_i^l = \mathbb{E}(m_i^l)$, where the expectation is over the randomness of the neighbors that send incorrect messages. We make the following assumption: $m_i^l$ is independent of $\{d_i\}$, $\{y_i\}$ and $\{\mathbf{f}_i^0\}$.

**Theorem 3.** *[Signed Messages] With the independence assumptions and under the "non-swapping condition", by allowing the messages to be optionally multiplied by a negative sign, in the initial stage, the movements of representations will **be less affected** by the initial homophily level $h_i$, and will be dependent on the **error rate** $e_i^l$. Specifically, the multiplicative factor $\gamma_i^1$ at the first layer is:*

$$\mathbb{E}_{\mathbf{A}, \{y_i\}, \{\mathbf{f}_i^{(0)}\}|d_i, v_i \in \mathcal{V}_1}(\mathbf{f}_i^{(1)}|d_i, v_i \in \mathcal{V}_1) = \left(\frac{(1-2e_i^0)(\rho + (1-\rho)h_i)d_i\bar{r}_i + 1}{d_i + 1}\right)\boldsymbol{\mu} \equiv \gamma_i^1\mathbb{E}(\mathbf{f}_i^{(0)}),$$

$$(4)$$

$$where \ \gamma_i^1 \in \begin{cases} (-\infty, \frac{1}{2}], & if \ e_i^0 \geq 0.5 \\ (0, 1], & if \ e_i^0 < 0.5 \ \& \ \bar{r}_i \leq \frac{1}{(1-2e_i^0)(\rho+(1-\rho)h_i)} \\ (1, \infty), & otherwise. \end{cases} \quad (5)$$

*Proof.* The proof is provided in App. A.5. $\square$

From Eq. 5, we see that when using signed messages, to benefit from case 3 ($\gamma_i^1 > 1$), the minimum relative degree satisfies: $\bar{r}_i > \frac{1}{(1-2e_i^0)(\rho+(1-\rho)h_i)}$. Given $h_i \leq 1$, if the error rate is low ($e_i^0 \ll 0.5$), we get: $\frac{1}{(1-2e_i^0)(\rho+(1-\rho)h_i)} \leq \frac{1}{(1+\rho)h_i-\rho}$, and $\frac{1}{(1+\rho)h_i-\rho}$ is the minimum relative degree required when not using signed messages. This implies that more nodes can benefit from using signed messages. We note that if low error rate cannot be guaranteed, signed messages may hurt the performance.

## 4 MODEL DESIGN

Based on our theoretical analysis, we propose two new, simple mechanisms to address both the heterophily and oversmoothing problems: signed messages and degree corrections. We integrate these mechanisms, along with a decaying combination of the current and previous node representations (Chen et al., 2020), into a generalized GCN model, GGCN. In § 5, we empirically show its effectiveness in addressing the two closely-related problems.

### 4.1 SIGNED MESSAGES

Thm 3 points out the importance of signed messages in tackling the heterophily and oversmoothing problems. Our first proposed mechanism uses cosine similarity to send signed messages.

For expressiveness, as in GCN (Kipf & Welling, 2016), we first perform a learnable linear transformation of each node's representation at the $l$-th layer: $\widehat{\mathbf{F}^{(l)}} = \mathbf{F}^{(l)}\mathbf{W}^{(l)} + \mathbf{b}^{(l)}$ Then, we define a sign function to be multiplied with the messages exchanged between neighbors. To allow for backpropagation of the gradient information, we approximate the sign function with cosine similarity. Denote $\mathbf{S}^l$ as the matrix which stores the sign information about the edges, defined as: $\mathbf{S}^{(l)}[i,j] = \texttt{Cosine}(\mathbf{f}_i^{(l)}, \mathbf{f}_j^{(l)})$ if $(i \neq j)$ & $(v_j \in \mathcal{N}_i)$; 0 otherwise.

In order to separate the contribution of similar neighbors (likely in the same class) from that of dissimilar neighbors (unlikely to be in the same class), we split $\mathbf{S}^{(l)}$ into a positive matrix $\mathbf{S}_{\text{pos}}^{(l)}$ and a negative matrix $\mathbf{S}_{\text{neg}}^{(l)}$. Thus, our proposed GGCN model learns a weighted combination of the self-representations, the positive messages, and the negative messages:

$$\mathbf{F}^{(l+1)} = \sigma\left(\hat{\alpha}^l(\hat{\beta}_0^l \, \widehat{\mathbf{F}^{(l)}} + \hat{\beta}_1^l(\mathbf{S}_{\text{pos}}^{(l)} \odot \tilde{\mathbf{A}})\widehat{\mathbf{F}^{(l)}} + \hat{\beta}_2^l(\mathbf{S}_{\text{neg}}^{(l)} \odot \tilde{\mathbf{A}})\widehat{\mathbf{F}^{(l)}})\right), \tag{6}$$

where $\hat{\beta}_0^l$, $\hat{\beta}_1^l$ and $\hat{\beta}_2^l$ are scalars obtained by applying softmax to the learned scalars $\beta_0^l$, $\beta_1^l$ and $\beta_2^l$; the non-negative scaling factor $\hat{\alpha}^l = \text{softplus}(\alpha^l)$ is derived from the learned scalar $\alpha^l$; $\odot$ is element-wise multiplication; and $\sigma$ is the nonlinear function Elu. We note that we learn *different* $\alpha$ and $\beta$ parameters *per layer* for flexibility. We also require the combined weights, $\hat{\alpha}^l\hat{\beta}_x^l$, to be non-negative so that they do not negate the intended effect of the signed information.

### 4.2 DEGREE CORRECTIONS

Our analysis in § 3.3 and § 3.4 highlights that, when the error rate and homophily level are high, oversmoothing is initially triggered by low-degree nodes. Thus, our second proposed mechanism aims to compensate for low degrees via degree corrections.

Based on Eq. (5), and given that most well-trained graph models have a relatively low error rate at the shallow layers (i.e, $e_i^l \ll 0.5$ for small $l$), we require that the node degrees satisfy $\overline{r_i} > \frac{1}{(1-2e_i^0)(\pi+(1-\pi)h_i)}$ to prevent oversmoothing. Since the node degrees cannot be modified, our strategy is to *rescale* them. Specifically, we correct the degrees by multiplying with scalars $\tau_{ij}^l$ and change the original formulation as follows:

$$(\tilde{A}\widehat{\mathbf{F}^{(l)}})[i,:] = \frac{\widehat{\mathbf{F}^{(l)}}[i,:]}{d_i + 1} + \boxed{\sum_{v_j \in \mathcal{N}_i} \frac{\widehat{\mathbf{F}^{(l)}}[j,:]}{\sqrt{d_i+1}\sqrt{d_j+1}}} \implies \sum_{v_j \in \mathcal{N}_i} \frac{\tau_{i,j}^l \, \widehat{\mathbf{F}^{(l)}}[j,:]}{\sqrt{d_i+1}\sqrt{d_j+1}}. \tag{7}$$

This multiplication is equivalent to changing the ratio $r_{ij}$ in Thm. 1 to $\sqrt{\frac{(\tau_{ij}^l)^2(d_i+1)}{d_j+1}}$. That is, a larger $\tau_{ij}^l$ increases the effective $\overline{r_i}$ at layer $l$. Training independent $\tau_{i,j}^l$ is not practical because it would require $O(|\mathcal{V}|^2)$ additional parameters per layer, which can lead to overfitting. Moreover, low-rank parameterizations suffer from unstable training dynamics. Intuitively, when $r_{ij}$ is small, we would like to compensate for it via a larger $\tau_{i,j}^l$. Thus, we set $\tau_{i,j}^l$ to be a function of $r_{ij}$ as follows:

$$\tau_{ij}^l = \text{softplus}\left(\lambda_0^l\left(\frac{1}{r_{ij}} - 1\right) + \lambda_1^l\right), \tag{8}$$

where $\lambda_0^l$ and $\lambda_1^l$ are learnable parameters. We subtract 1 so that when $r_{ij} = 1$ (i.e., $d_i = d_j$), then $\tau_{ij}^l = \text{softplus}(\lambda_1^l)$ is a constant bias.

In our proposed GGCN model, we combine the signed messages in Eq. (6) and our degree correction mechanism to obtain the representations at layer $l+1$:

$$\mathbf{F}^{(l+1)} = \sigma\left(\hat{\alpha}^l\left(\hat{\beta}_0^l\widehat{\mathbf{F}^{(l)}} + \hat{\beta}_1^l(\mathbf{S}_{pos}^{(l)} \odot \tilde{\mathbf{A}} \odot \boldsymbol{\mathcal{T}}^{(l)})\widehat{\mathbf{F}^{(l)}} + \hat{\beta}_2^l(\mathbf{S}_{neg}^{(l)} \odot \tilde{\mathbf{A}} \odot \boldsymbol{\mathcal{T}}^{(l)})\widehat{\mathbf{F}^{(l)}}\right)\right), \tag{9}$$

where $\boldsymbol{\mathcal{T}}^{(l)}$ is a matrix with elements $\tau_{ij}^l$.

### 4.3 DECAYING AGGREGATION

In addition to our two proposed mechanisms that are theoretically grounded in our analysis (§ 3), we also incorporate into GGCN an existing design—decaying aggregation of messages—that empirically increases performance. However, we note that, even without this design, our GCN architecture still performs well under heterophily and is robust to oversmoothing (App. §B.1).

Decaying aggregation was introduced in (Chen et al., 2020) as a way to slow down the convergence rate of node representations. Inspired by this work, we modify the decaying function, $\hat{\eta}$, and integrate it to our GGCN model:

$$\mathbf{F}^{(l+1)} = \boxed{\mathbf{F}^{(l)} + \hat{\eta}} \left( \sigma \left( \hat{\alpha}^l (\hat{\beta}_0^l \widehat{\mathbf{F}^{(l)}} + \hat{\beta}_1^l (\mathbf{S}_{\text{pos}}^{(l)} \odot \tilde{\mathbf{A}} \odot \boldsymbol{\mathcal{T}}^{(l)}) \widehat{\mathbf{F}^{(l)}} + \hat{\beta}_2^l (\mathbf{S}_{\text{neg}}^{(l)} \odot \tilde{\mathbf{A}} \odot \boldsymbol{\mathcal{T}}^{(l)}) \widehat{\mathbf{F}^{(l)}}) \right) \right), \quad (10)$$

In practice, we found that the following decaying function works well: $\hat{\eta} \equiv \ln(\frac{\eta}{l^k} + 1)$, iff $l \geq l_0; \hat{\eta} = 1$, otherwise. The hyperparameters $k, l_0, \eta$ are tuned on the validation set.

## 5 EXPERIMENTS

Due to the space limit, in the main paper, we focus on the following three questions: (**Q1**) Compared to the baselines, how does GGCN perform on homophilous and heterophilous graphs? (**Q2**) How robust is it against oversmoothing under homophily and heterophily? (**Q3**) How can we verify the correctness of our theorems on real datasets? We provide the ablation study of signed messages and degree correction in App. § B.1 and the study of different normalization strategies in App. § B.2.

### 5.1 EXPERIMENTAL SETUP

**Datasets.** We evaluate the performance of our GGCN model and existing GNNs in node classification on various real-world datasets (Tang et al., 2009; Rozemberczki et al., 2019; Sen et al., 2008; Namata et al., 2012; Bojchevski & Günnemann, 2018; Shchur et al., 2018). We provide their summary statistics in Table 1, where we compute the homophily level $h$ of a graph as the average of $h_i$ of all nodes $v_i \in \mathcal{V}$. For all benchmarks, we use the feature vectors, class labels, and 10 random splits (48%/32%/20% of nodes per class for train/validation/test[1]) from (Pei et al., 2019).

**Baselines.** For baselines we use (**1**) classic GNN models for node classification: vanilla GCN (Kipf & Welling, 2016), GAT (Veličković et al., 2017) and GraphSage (Hamilton et al., 2017); (**2**) recent models tackling heterophily: Geom-GCN (Pei et al., 2019), H2GCN (Zhu et al., 2020) and GPRGNN (Chien et al., 2021); (**3**) models tackling oversmoothing: PairNorm (Zhao & Akoglu, 2019) and GCNII (Chen et al., 2020) (state-of-the-art); and (**4**) 2-layer MLP (with dropout and Elu non-linearity). For GCN, PairNorm, Geom-GCN, GCNII, H2GCN and GPRGNN, we use the original codes provided by the authors. For GAT, we use the code from a well-accepted Github repository[2]. For GraphSage, we report the results from (Zhu et al., 2020), which uses the same data and splits. For the baselines that have multiple variants (Geom-GCN, GCNII, H2GCN), we choose the best variant for each dataset and denote them as [model]*. For each dataset and baseline, if applicable, we use the best hyperparameters provided by the authors. Otherwise, we perform parameter search to set the best hyperparameters for each baseline. The setting and analysis of hyperparameters are in App. § C.

**Machine.** We ran our experiments on Nvidia V100 GPU.

Table 1: Real data: mean accuracy ± stdev over different data splits. Per GNN model, we report the best performance across different layers. Best model per benchmark highlighted in gray. The "†" results (GraphSAGE) are obtained from (Zhu et al., 2020).

| | Texas | Wisconsin | Actor | Squirrel | Chameleon | Cornell | Citeseer | Pubmed | Cora | Avg Rank |
|---|---|---|---|---|---|---|---|---|---|---|
| Hom. level $h$ | 0.11 | 0.21 | 0.22 | 0.22 | 0.23 | 0.3 | 0.74 | 0.8 | 0.81 | |
| #Nodes | 183 | 251 | 7,600 | 5,201 | 2,277 | 183 | 3,327 | 19,717 | 2,708 | |
| #Edges | 295 | 466 | 26,752 | 198,493 | 31,421 | 280 | 4,676 | 44,327 | 5,278 | |
| #Classes | 5 | 5 | 5 | 5 | 5 | 5 | 7 | 3 | 6 | |
| GGCN (ours) | 84.86±4.55 | 86.86±3.29 | 37.54±1.56 | 55.17±1.58 | 71.14±1.84 | 85.68±6.63 | 77.14±1.45 | 89.15±0.37 | 87.95±1.05 | 1.78 |
| GPRGNN | 78.38±4.36 | 82.94±4.21 | 34.63±1.22 | 31.61±1.24 | 46.58±1.71 | 80.27±8.11 | 77.13±1.67 | 87.54±0.38 | 87.95±1.18 | 5.56 |
| H2GCN* | 84.86±7.23 | 87.65±4.98 | 35.70±1.00 | 36.48±1.86 | 60.11±2.15 | 82.70±5.28 | 77.11±1.57 | 89.49±0.38 | 87.87±1.20 | 3.89 |
| GCNII* | 77.57±3.83 | 80.39±3.4 | 37.44±1.30 | 38.47±1.58 | 63.86±3.04 | 77.86±3.79 | 77.33±1.48 | 90.15±0.43 | 88.37±1.25 | 3.56 |
| Geom-GCN* | 66.76±2.72 | 64.51±3.66 | 31.59±1.15 | 38.15±0.92 | 60.00±2.81 | 60.54±3.67 | 78.02±1.15 | 89.95±0.47 | 85.35±1.57 | 6.11 |
| PairNorm | 60.27±4.34 | 48.43±6.14 | 27.40±1.24 | 50.44±2.04 | 62.74±2.82 | 58.92±3.15 | 73.59±1.47 | 87.53±0.44 | 85.79±1.01 | 7.78 |
| GraphSAGE† | 82.43±6.14 | 81.18±5.56 | 34.23±0.99 | 41.61±0.74 | 58.73±1.68 | 75.95±5.01 | 76.04±1.30 | 88.45±0.50 | 86.90±1.04 | 5.78 |
| GCN | 55.14±5.16 | 51.76±3.06 | 27.32±1.10 | 53.43±2.01 | 64.82±2.24 | 60.54±5.3 | 76.50±1.36 | 88.42±0.5 | 86.98±1.27 | 6.56 |
| GAT | 52.16±6.63 | 49.41±4.09 | 27.44±0.89 | 40.72±1.55 | 60.26±2.5 | 61.89±5.05 | 76.55±1.23 | 86.33±0.48 | 87.30±1.10 | 7.22 |
| MLP | 80.81±4.75 | 85.29±3.31 | 36.53±0.70 | 28.77±1.56 | 46.21±2.99 | 81.89±6.40 | 74.02±1.90 | 87.16±0.37 | 75.69±2.00 | 6.78 |

[1](Pei et al., 2019) claims that the ratios are 60%/20%/20%, which is different from the actual data splits shared on GitHub.

[2]https://github.com/Diego999/pyGAT

Table 2: Model performance for different layers: mean accuracy ± stdev over different data splits. Per dataset and GNN model, we also report the layer at which the best performance (given in Table 1) is achieved. 'OOM': out of memory; 'INS': numerical instability. For larger font, refer to Table D.1 in the Appendix.

| Layers | 2 | 4 | 8 | 16 | 32 | 64 | Best | 2 | 4 | 8 | 16 | 32 | 64 | Best |
|---|---|---|---|---|---|---|---|---|---|---|---|---|---|---|
| | **Cora** (h=0.81) | | | | | | | **Citeseer** (h=0.74) | | | | | | |
| GGCN (ours) | $87.00_{\pm1.15}$ | $87.48_{\pm1.32}$ | $87.63_{\pm1.33}$ | $87.51_{\pm1.19}$ | $87.95_{\pm1.05}$ | $87.28_{\pm1.41}$ | 32 | $76.83_{\pm1.82}$ | $76.77_{\pm1.48}$ | $76.91_{\pm1.56}$ | $76.88_{\pm1.56}$ | $76.97_{\pm1.52}$ | $76.65_{\pm1.38}$ | 10 |
| GPRGNN | $87.93_{\pm1.11}$ | $87.95_{\pm1.18}$ | $87.87_{\pm1.41}$ | $87.26_{\pm1.51}$ | $87.18_{\pm1.29}$ | $87.32_{\pm1.21}$ | 4 | $77.13_{\pm1.67}$ | $77.05_{\pm1.43}$ | $77.09_{\pm1.62}$ | $76.00_{\pm1.64}$ | $74.97_{\pm1.47}$ | $74.41_{\pm1.65}$ | 2 |
| H2GCN* | $87.87_{\pm1.20}$ | $86.10_{\pm1.51}$ | $86.18_{\pm2.10}$ | OOM | OOM | OOM | 2 | $76.90_{\pm1.80}$ | $76.09_{\pm1.54}$ | $74.10_{\pm1.83}$ | OOM | OOM | OOM | 1 |
| GCNII* | $85.35_{\pm1.56}$ | $85.35_{\pm1.48}$ | $86.38_{\pm0.98}$ | $87.12_{\pm1.11}$ | $87.95_{\pm1.23}$ | $88.37_{\pm1.25}$ | 64 | $75.42_{\pm1.78}$ | $75.29_{\pm1.90}$ | $76.00_{\pm1.66}$ | $76.96_{\pm1.38}$ | $77.33_{\pm1.48}$ | $77.18_{\pm1.47}$ | 32 |
| PairNorm | $85.79_{\pm1.01}$ | $85.07_{\pm0.91}$ | $84.65_{\pm1.09}$ | $82.21_{\pm2.84}$ | $60.32_{\pm8.28}$ | $44.39_{\pm5.60}$ | 2 | $73.59_{\pm1.47}$ | $72.62_{\pm1.97}$ | $72.32_{\pm1.58}$ | $59.71_{\pm15.97}$ | $27.21_{\pm10.95}$ | $23.82_{\pm6.64}$ | 2 |
| Geom-GCN* | $85.35_{\pm1.57}$ | $21.01_{\pm2.61}$ | $13.98_{\pm1.48}$ | $13.98_{\pm1.48}$ | $13.98_{\pm1.48}$ | $13.98_{\pm1.48}$ | 2 | $78.02_{\pm1.15}$ | $23.01_{\pm1.95}$ | $7.23_{\pm0.87}$ | $7.23_{\pm0.87}$ | $7.23_{\pm0.87}$ | $7.23_{\pm0.87}$ | 2 |
| GCN | $86.98_{\pm1.27}$ | $83.24_{\pm1.56}$ | $31.03_{\pm3.08}$ | $31.05_{\pm2.36}$ | $30.76_{\pm3.43}$ | $31.89_{\pm2.08}$ | 2 | $76.50_{\pm1.36}$ | $64.33_{\pm8.27}$ | $24.18_{\pm1.71}$ | $23.07_{\pm2.95}$ | $25.3_{\pm1.77}$ | $24.73_{\pm1.66}$ | 2 |
| GAT | $87.30_{\pm1.10}$ | $86.50_{\pm1.20}$ | $84.97_{\pm1.24}$ | INS | INS | INS | 2 | $76.55_{\pm1.23}$ | $75.33_{\pm1.39}$ | $66.57_{\pm5.08}$ | INS | INS | INS | 2 |
| | **Cornell** (h=0.3) | | | | | | | **Chameleon** (h=0.23) | | | | | | |
| GGCN (ours) | $83.78_{\pm6.73}$ | $83.78_{\pm6.16}$ | $84.86_{\pm5.69}$ | $83.78_{\pm6.73}$ | $83.78_{\pm6.51}$ | $84.32_{\pm5.90}$ | 6 | $70.77_{\pm1.42}$ | $69.58_{\pm2.68}$ | $70.33_{\pm1.70}$ | $70.44_{\pm1.82}$ | $70.29_{\pm1.62}$ | $70.20_{\pm1.95}$ | 5 |
| GPRGNN | $76.76_{\pm8.22}$ | $77.57_{\pm7.46}$ | $80.27_{\pm8.11}$ | $78.38_{\pm6.04}$ | $74.59_{\pm7.66}$ | $70.00_{\pm5.73}$ | 8 | $46.58_{\pm1.771}$ | $45.72_{\pm3.45}$ | $41.16_{\pm5.79}$ | $39.58_{\pm7.85}$ | $35.42_{\pm8.52}$ | $36.38_{\pm2.40}$ | 2 |
| H2GCN* | $81.89_{\pm5.98}$ | $82.70_{\pm6.27}$ | $80.27_{\pm6.63}$ | OOM | OOM | OOM | 1 | $59.06_{\pm1.85}$ | $60.11_{\pm2.15}$ | OOM | OOM | OOM | OOM | 4 |
| GCNII* | $67.57_{\pm11.34}$ | $64.59_{\pm9.63}$ | $73.24_{\pm5.91}$ | $77.84_{\pm3.97}$ | $75.41_{\pm5.47}$ | $73.78_{\pm4.37}$ | 16 | $61.07_{\pm4.10}$ | $63.86_{\pm3.04}$ | $62.89_{\pm1.18}$ | $60.20_{\pm2.10}$ | $56.97_{\pm1.81}$ | $55.99_{\pm2.27}$ | 4 |
| PairNorm | $50.27_{\pm7.17}$ | $53.51_{\pm8.00}$ | $58.38_{\pm3.01}$ | $58.38_{\pm3.01}$ | $58.92_{\pm3.15}$ | $58.92_{\pm3.15}$ | 32 | $62.74_{\pm2.82}$ | $59.01_{\pm2.80}$ | $54.12_{\pm2.24}$ | $46.38_{\pm2.23}$ | $46.78_{\pm2.26}$ | $46.27_{\pm3.24}$ | 2 |
| Geom-GCN* | $60.54_{\pm3.67}$ | $23.78_{\pm11.64}$ | $12.97_{\pm2.91}$ | $12.97_{\pm2.91}$ | $12.97_{\pm2.91}$ | $12.97_{\pm2.91}$ | 2 | $60.00_{\pm2.81}$ | $19.17_{\pm1.66}$ | $19.58_{\pm1.73}$ | $19.58_{\pm1.73}$ | $19.58_{\pm1.73}$ | $19.58_{\pm1.73}$ | 2 |
| GCN | $60.54_{\pm5.30}$ | $59.19_{\pm3.30}$ | $58.92_{\pm3.15}$ | $58.92_{\pm3.15}$ | $58.92_{\pm3.15}$ | $58.92_{\pm3.15}$ | 2 | $64.82_{\pm2.24}$ | $53.11_{\pm4.44}$ | $35.15_{\pm3.14}$ | $35.39_{\pm3.23}$ | $35.20_{\pm3.25}$ | $35.50_{\pm3.08}$ | 2 |
| GAT | $61.89_{\pm5.05}$ | $58.38_{\pm4.05}$ | $58.38_{\pm3.86}$ | INS | INS | INS | 2 | $60.26_{\pm2.50}$ | $48.71_{\pm2.96}$ | $35.09_{\pm3.55}$ | INS | INS | INS | 2 |

## 5.2 Q1. Performance Under Homophily & Heterophily

Table 1 provides the test accuracy of different GNNs on the supervised node classification task over datasets with varying homophily levels (arranged from low homophily to high homophily). A graph's homophily level is the average of nodes' homophily levels. We report the best performance of each model across different layers. In App. § B.4, we provide the analysis based on the nodes' homophily levels (§ 3), which is a better metric to predict GCN's performance and is aligned with our theory.

GGCN performs the best in terms of average rank (1.78) across all datasets, which suggests its strong adaptability to graphs of various homophily levels. In particular, GGCN achieves the highest accuracy in 5 out of 6 heterophilous graphs ($h$ is low). For datasets like Chameleon and Cornell, GGCN enhances accuracy by around 6% and 3% compared to the second-best model. On homophily datasets (Citeseer, Pubmed, Cora), the accuracy of GGCN is within a 1% difference of the best model.

Our experiments highlight that MLP is a good baseline in heterophilous datasets. In heterophilous graphs, the models that are not specifically designed for heterophily usually perform worse than an MLP. Though H2GCN* is the second best model in heterophilous datasets, we can still see that in the Actor dataset, MLP performs better. GPRGNN and Geom-GCN*, which are specifically designed for heterophily, achieve better performance than classic GNNs (GCN and GAT) in heterophilous datasets, but do not show clear advantage over MLP. Our GGCN model is the only model that performs better than MLP across all the datasets.

In general, GNN models perform well in homophilous datasets. GCNII* performs the best, and GGCN, H2GCN*, GPRGNN and Geom-GCN* also achieve high performance.

## 5.3 Q2. Oversmoothing

We also test how robust the models are to oversmoothing. To this end, we measure the supervised node classification accuracy for 2 to 64 layers. Table 2 presents the results for two homophilous datasets (top) and two heterophilous datasets (bottom). Per model, we also report the layer at which the best performance is achieved (column 'Best'). We provides Table 2 in larger font in App. § D.

According to Table 2, GGCN and GCNII* achieve **increase** in accuracy when stacking more layers in four datasets, while GPRGNN and PairNorm exhibit robustness against oversmoothing. Models that are not designed for oversmoothing have various issues. The performance of GCN and Geom-GCN* drops rapidly as the number of layers grows; H2GCN* requires concatenating all the intermediate outputs and quickly reaches memory capacity; GAT's attention mechanism also has high memory requirements. We also find that GAT needs careful initialization when stacking many layers as it may suffer from numerical instability in sparse tensor operations.

In general, models like GGCN, GCNII*, and GPRGNN that perform well under heterophily usually exhibit higher resilience against oversmoothing. One exception is Geom-GCN*, which suffers more than GCN. This model incorporates structurally similar nodes into each node's neighborhood; this design may benefit Geom-GCN* in the shallow layers as the node degrees increase. However, as we point out in Thm. 2, when the effective homophily is low, higher degrees are harmful. If the structurally similar nodes introduce lower homophily levels, their performance will rapidly degrade once the effective homophily is lower than $\frac{\rho_i^l}{1+\rho_i^l}$. On the other hand, GGCN *virtually* changes the degrees thanks to the degree correction mechanism (§ 4.2), and, in practice, this design has **positive** impact on its robustness to oversmoothing.

### 5.4 Q3. EMPIRICAL VERIFICATION OF THE INITIAL & DEVELOPING STAGES

Using the vanilla GCN model (Kipf & Welling, 2016), we validate our theorems by measuring the test accuracy and effective homophily for different node degrees (binned logarithmically) on real datasets. We estimate the effective homophily as the portion of the same-class neighbors that are correctly classified *before* the last propagation. Figure 1 shows the results for Citeseer. In the initial stage ($\widehat{h}_i^l$ is high), the accuracy increases with the degree, but in the developing stage, the trend changes, with high-degree nodes being impacted the most, as predicted by our theorems. In App. § B.3, we provide more details for this analysis and we further verify our conclusion on Cora.

## 6 RELATED WORK

**Graph Convolutional Neural Networks.** Early on, (Defferrard et al., 2016) proposed a GCN model that combines spectral filtering of graph signals and non-linearity for supervised node classification. The scalability and numerical stability of GCNs was later improved with a localized first-order approximation of spectral graph convolutions proposed in (Kipf & Welling, 2016). (Veličković et al., 2017) proposes the first graph attention network to improve neighborhood aggregation. Many more GCN variants have been proposed for different applications such as: computer vision (Satorras & Estrach, 2018), social science (Li & Goldwasser, 2019), biology (Yan et al., 2019), algorithmic tasks (Veličković et al., 2020; Yan et al., 2020), and inductive classification (Hamilton et al., 2017). Concurrent work (Baranwal et al., 2021) provides theoretic analysis on the linear separability of graph convolution but does not provide effective strategies to increase the separability.

**Oversmoothing.** The oversmoothing problem was first discussed in (Li et al., 2018), which proved that by repeatedly applying Laplacian smoothing, the representations of nodes within each connected component of the graph converge to the same value. Since then, various empirical solutions have been proposed: residual connections and dilated convolutions (Li et al., 2019); skip links (Xu et al., 2018); new normalization strategies (Zhao & Akoglu, 2019); edge dropout (Rong et al., 2019); and a new model that even increases performance as more layers are stacked (Chen et al., 2020). Some recent works provide theoretical analyses: (Oono & Suzuki, 2019) showed that a $k$-layer renormalized graph convolution with a residual link simulates a lazy random walk and (Chen et al., 2020) proved that the convergence rate is related to the spectral gap of the graph.

**Heterophily & GCNs.** Heterophily has recently been recognized as an important issue for GCNs. It is first outlined in the context of GCNs in (Pei et al., 2019). (Zhu et al., 2020) identified a set of effective designs that allow GCNs to generalize to challenging heterophilous settings, and (Zhu et al., 2021) introduced a new GCN model that leverages ideas from belief propagation (Gatterbauer et al., 2015). Though recent work (Chen et al., 2020) focused on solving the oversmoothing problem, it also empirically showed improvement on heterophilous datasets; these empirical observations formed the basis of our work. Finally, (Chien et al., 2021) recently proposed a PageRank-based model that performs well under heterophily and alleviates the oversmoothing problem. However, they view the two problems independently and analyze their model via an asymptotic spectral perspective. Our work studies the representation dynamics and unveils the connections between the oversmoothing and heterophily problems theoretically and empirically. As we demonstrate with GGCN, addressing both issues in a principled manner provides superior performance across a variety of datasets.

## 7 CONCLUSION

Our work provides the first theoretical and empirical analysis that unveils the connections between the oversmoothing and heterophily problems. By analyzing the statistical change of the node representations after the graph convolution, we identified two causes, i.e., the relative degree of a node compared to its neighbors and the level of heterophily in its neighborhood, which influence the movements of node representations and lead to a higher misclassification rate. Based on our new, unified theoretical perspective, we obtained three important insights: (1) Nodes with high heterophily tend to be misclassified after graph convolution; (2) Even with low heterophily, low-degree nodes can trigger a pseudo-heterophily situation that explains oversmoothing. (3) Allowing signed messages (instead of only positive messages) helps alleviate the heterophily and oversmoothing problems. Based on these insights, we designed a generalized model, GGCN, that addresses the identified causes using signed messages and degree corrections. Though other designs may also address these two problems, our work points out two effective directions that are theoretically grounded (§ 3). In summary, our research suggests it is beneficial to study the oversmoothing and heterophily problems jointly; this leads to architectural insights that can improve the learned representations of graph neural network models across a variety of domains.

## 8 REPRODUCIBILITY STATEMENT

To reproduce our experimental results, we provide the references to the datasets and baselines in § 5. For the baselines, we use code provided by the authors and set the hyperparameters as the authors suggest. In App. § C, we provide detailed hyperparameter setting for GGCN. We also provide our code in the supplementary material and we will share the link to our code upon acceptance.

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

# A    DETAILED PROOFS OF THEOREMS IN § 3

## A.1    PROOF OF LEMMA 1

*Proof.* The loss function for SGC is: `CrossEntropy(Softmax(`$\mathbf{F}^{(L)}\mathbf{W}^{(L)} + \mathbf{b}^{(L)}$`),` $y_i$`)`. Rewrite $\mathbf{W}^{(L)}$ into an array of column vectors: $\mathbf{W}^{(L)} = [\mathbf{w}_0^{(L)}, \mathbf{w}_1^{(L)}, \dots \mathbf{w}_{|\mathcal{L}|}^{(L)}]$, $\mathbf{b}^{(L)}$ into an array of scalars: $\mathbf{b}^{(L)} = [b_0^{(L)}, b_1^{(L)}, \dots b_{|\mathcal{L}|}^{(L)}]$, and $\mathbf{F}^{(L)}$ into an array of row vectors: $\mathbf{F}^{(L)} = [\mathbf{f}_0^{(L)}; \mathbf{f}_1^{(L)}; \dots \mathbf{f}_{|\mathcal{V}|}^{(L)}]$ ($[\cdot; \cdot]$ means stacking vertically). Let $p_{i,c} \equiv \frac{e^{\mathbf{f}_i^{(L)}\mathbf{w}_c^{(L)}+b_c^{(L)}}}{\sum_j e^{\mathbf{f}_i^{(L)}\mathbf{w}_j^{(L)}+b_j^{(L)}}}$ and let $y_{i,c} \equiv \begin{cases} 1, & \text{iff } y_i = c \\ 0, & \text{otherwise} \end{cases}$ then we rewrite the loss function as:

$$-\frac{1}{n}\sum_{i=1}^{n}\sum_{c=1}^{|\mathcal{L}|} y_{i,c} \cdot \log(p_{i,c}), \tag{11}$$

$n$ is the number of nodes used for training. To predict that the node $v_i$ belongs to class $c$, we require: $\forall c' \neq c, p_{i,c} > p_{i,c'}$. Thus, we have:

$$\frac{e^{\mathbf{f}_i^{(L)}\mathbf{w}_c^{(L)}+b_c^{(L)}}}{\sum_j e^{\mathbf{f}_i^{(L)}\mathbf{w}_j^{(L)}+b_j^{(L)}}} > \frac{e^{\mathbf{f}_i^{(L)}\mathbf{w}_{c'}^{(L)}+b_{c'}^{(L)}}}{\sum_j e^{\mathbf{f}_i^{(L)}\mathbf{w}_j^{(L)}+b_j^{(L)}}}. \tag{12}$$

It is equivalent to:

$$\mathbf{f}_i^{(L)}\mathbf{w}_c^{(L)} + b_c^{(L)} > \mathbf{f}_i^{(L)}\mathbf{w}_{c'}^{(L)} + b_{c'}^{(L)}. \tag{13}$$

It shows that the decision region of class $c$ is decided by a set of hyper-planes:

$$\mathbf{x}(\mathbf{w}_c^{(L)} - \mathbf{w}_{c'}^{(L)}) + (b_c^{(L)} - b_{c'}^{(L)}) > 0. \tag{14}$$

$\square$

$\mathbf{x}$ is the vector variable that describes the plane. Any vectors that satisfy the equation are on this plane. We note that for binary classification, there is one hyper-plane.

## A.2    PROOF OF LEMMA 2

Before giving the proof, we first explain the meaning of the Lemma. As we point out in § 3, the misclassification rate of an $L$-layer SGC can be studied through its last logistic-regression layer, the input to which is $\{\mathbf{f}_i^{(L)}\}$. To study the misclassification rate of an $(L+1)$-layer SGC, we can view the input to the last layer $\{\mathbf{f}_i^{(L+1)}\}$ as being moved from $\{\mathbf{f}_i^{(L)}\}$. The misclassification rate is closely related to the decision boundary and will be our tool for studying the change of the misclassification rate. This Lemma studies the movements that bring the node representations closer to the original decision boundary; we will prove that moving towards the original decision boundary by a non-zero step is not beneficial (i.e., harmful) to the SGC's performance.

*Proof.* We prove it by contradiction and suppose that after the movements, the total misclassification rate is lowered. We denote the conditional distribution of $\{\mathbf{f}_i^{(L)}\}$ as $f_1^L(\mathbf{x})$ conditioned on that they are from class 1 and $f_2^L(\mathbf{x})$ conditioned on that they are from class 1 or class 2, respectively. To note that, the distributions of different layers are different.

After the movements, the representations from class 1 become: $\{\mathbf{f}_i^{(L)} - t\mathbf{w}^{*T}\}, t > 0$. $t\mathbf{w}^{*T}$ represents moving towards the original decision boundary along the norm direction by a non-zero step. This causes a corresponding change in the conditional PDF given that they are from class 1: $f_1^L(\mathbf{x} + t\mathbf{w}^{*T})$.

Recall that in § 3, the parameters for the original optimal hyperplane $\mathbf{w}^*$ and $b^*$, and the parameters for the later optimal hyperplane $\mathbf{w}^{*\prime}$ and $b^{*\prime}$ are normalized such that when $\mathbf{x}\mathbf{w}^* + b^* > 0$ and $\mathbf{x}\mathbf{w}^{*\prime} + b^{*\prime} > 0$, we predict class 1.

The new misclassification rate conditioned on class 1 is:

$$M_t' = \int_{\mathbf{x}\mathbf{w}^{*\prime}+b^{*\prime}<0} f_1^L(\mathbf{x} + t\mathbf{w}^{*T})d\mathbf{x} = \int_{(\mathbf{x}-t\mathbf{w}^{*T})\mathbf{w}^{*\prime}+b^{*\prime}<0} f_1^L(\mathbf{x})d\mathbf{x}. \tag{15}$$

The new misclassification rate conditioned on class 2 is:

$$M'_r = \int_{\mathbf{x}\mathbf{w}^{*\prime}+b^{*\prime}>0} f_2^L(\mathbf{x})d\mathbf{x}. \tag{16}$$

The new total misclassification rate is:

$$
\begin{aligned}
M' &= M'_t \mathbb{P}(v_i \in \mathcal{V}_1) + M'_r \mathbb{P}(v_i \in \mathcal{V}_2) \\
&= M'_t \frac{\mathbb{P}(v_i \in \mathcal{V}_1)}{\mathbb{P}(v_i \in \mathcal{V}_1) + \mathbb{P}(v_i \in \mathcal{V}_2)} + M'_r \frac{\mathbb{P}(v_i \in \mathcal{V}_2)}{\mathbb{P}(v_i \in \mathcal{V}_1) + \mathbb{P}(v_i \in \mathcal{V}_2)} \\
&= \frac{\rho}{\rho+1} M'_t + \frac{1}{\rho+1} M'_r.
\end{aligned}
\tag{17}
$$

Next, we will prove that if the total misclassification is lowered, $\mathbf{x}\mathbf{w}^* + b^* = 0$ is not the optimal hyper-plane before the movements which should achieve the lowest total misclassification rate.

Consider a hyper-plane $\mathbf{x}\mathbf{w}^{*\prime} + b^{*\prime} - \frac{t\mathbf{w}^{*T}\mathbf{w}^{*\prime}}{2} = 0$. Given that $\mathbf{w}^{*T}\mathbf{w}^{*\prime} > 0$ and $t > 0$, for $\forall \mathbf{x}$, s.t. $\mathbf{x}\mathbf{w}^{*\prime}+b^{*\prime} - \frac{t\mathbf{w}^{*T}\mathbf{w}^{*\prime}}{2} < 0$, we have: $(\mathbf{x}-t\mathbf{w}^{*T})\mathbf{w}^{*\prime}+b^{*\prime} = \mathbf{x}\mathbf{w}^{*\prime}+b^{*\prime} - \frac{t\mathbf{w}^{*T}\mathbf{w}^{*\prime}}{2} - \frac{t\mathbf{w}^{*T}\mathbf{w}^{*\prime}}{2} < 0$. It means:

$$\{\mathbf{x}|(\mathbf{x} - t\mathbf{w}^{*T})\mathbf{w}^{*\prime} + b^{*\prime} < 0\} \supseteq \{\mathbf{x}|\mathbf{x}\mathbf{w}^{*\prime} + b^{*\prime} - \frac{t\mathbf{w}^{*T}\mathbf{w}^{*\prime}}{2} < 0\}. \tag{18}$$

Because $f_1^L(\mathbf{x})$ is a PDF which is a nonnegative function,

$$\int_{(\mathbf{x}-t\mathbf{w}^{*T})\mathbf{w}^{*\prime}+b^{*\prime}<0} f_1^L(\mathbf{x})d\mathbf{x} \geq \int_{\mathbf{x}\mathbf{w}^{*\prime}+b^{*\prime}-\frac{t\mathbf{w}^{*T}\mathbf{w}^{*\prime}}{2}<0} f_1^L(\mathbf{x})d\mathbf{x}. \tag{19}$$

Similarly, we can obtain

$$\{\mathbf{x}|\mathbf{x}\mathbf{w}^{*\prime} + b^{*\prime} > 0\} \supseteq \{\mathbf{x}|\mathbf{x}\mathbf{w}^{*\prime} + b^{*\prime} - \frac{t\mathbf{w}^{*T}\mathbf{w}^{*\prime}}{2} > 0\}. \tag{20}$$

Thus,

$$\int_{\mathbf{x}\mathbf{w}^{*\prime}+b^{*\prime}>0} f_2^L(\mathbf{x})d\mathbf{x} \geq \int_{\mathbf{x}\mathbf{w}^{*\prime}+b^{*\prime}-\frac{t\mathbf{w}^{*T}\mathbf{w}^{*\prime}}{2}>0} f_2^L(\mathbf{x})d\mathbf{x}. \tag{21}$$

Define

$$M_f \equiv \frac{\rho}{\rho+1} \int_{\mathbf{x}\mathbf{w}^{*\prime}+b^{*\prime}-\frac{t\mathbf{w}^{*T}\mathbf{w}^{*\prime}}{2}<0} f_1^L(\mathbf{x})d\mathbf{x} + \frac{1}{\rho+1} \int_{\mathbf{x}\mathbf{w}^{*\prime}+b^{*\prime}-\frac{t\mathbf{w}^{*T}\mathbf{w}^{*\prime}}{2}>0} f_2^L(\mathbf{x})d\mathbf{x}. \tag{22}$$

The quantity $M_f$ represents the total misclassification rate before the movements if the decision boundary is $\mathbf{x}\mathbf{w}^{*\prime} + b^{*\prime} - \frac{t\mathbf{w}^{*T}\mathbf{w}^{*\prime}}{2} = 0$. Given Eq. 17, 19 and 21, we have:

$$M' \geq M_f. \tag{23}$$

Let $M$ denotes the total classification rate before the movements. Based on the assumption that the total misclassification rate is lowered after the movements ($M > M'$), we have:

$$M > M_f. \tag{24}$$

Eq. 24 indicates that we find a hyper-plane $\mathbf{x}\mathbf{w}^{*\prime} + b^{*\prime} - \frac{t\mathbf{w}^{*T}\mathbf{w}^{*\prime}}{2} = 0$, which yields smaller total misclassification rate than $\mathbf{x}\mathbf{w}^* + b^* = 0$ before the movements. This contradicts to the fact that $\mathbf{x}\mathbf{w}^* + b^* = 0$ is the optimal hyper-plane before the movements. $\square$

### A.3 PROOF OF THEOREM 1

*Proof.* The first layer node representations are given by:

$$\mathbf{f}_i^{(1)} = \frac{\mathbf{f}_i^{(0)}}{d_i + 1} + \sum_{j \in \mathcal{N}_i} \frac{\mathbf{f}_j^{(0)}}{\sqrt{d_i + 1} \cdot \sqrt{d_j + 1}}. \tag{25}$$

Without loss of generality, we assume node $v_i$ is in the first class $\mathcal{V}_1$. Then, we can express the conditional expectation of $\mathbf{f}_i^{(1)}$ as:

$$\mathbb{E}_{\mathbf{A},\{y_i\},\{\mathbf{f}_i^{(0)}\}|d_i,v_i\in\mathcal{V}_1}(\mathbf{f}_i^{(1)}|v_i\in\mathcal{V}_1,d_i) = \mathbb{E}_{\mathbf{A}|d_i,v_i\in\mathcal{V}_1}(\mathbb{E}_{\{y_i\},\{\mathbf{f}_i^{(0)}\}|\mathbf{A},v_i\in\mathcal{V}_1}(\mathbf{f}_i^{(1)}|v_i\in\mathcal{V}_1,\mathbf{A})) \tag{26}$$

Recall that $\mathbf{A}$ is the graph adjacency matrix; the first expectation is taken over the randomness of ground truth labels and initial input features, and the second expectation is taken over the randomness of graph structure ($\mathbf{A}$) given the degree of node $v_i$ and its label. Denote $k_i = \mathbb{E}_{\{y_i\},\{\mathbf{f}_i^{(0)}\}|\mathbf{A},v_i\in\mathcal{V}_1}(\frac{|\mathcal{N}_i^s|}{|\mathcal{N}_i|})$ and $h_i = \mathbb{E}_{\mathbf{A},\{y_i\},\{\mathbf{f}_i^{(0)}\}|v_i\in\mathcal{V}_1,d_i}(\frac{|\mathcal{N}_i^s|}{|\mathcal{N}_i|})$.

$$\mathbb{E}_{\{y_i\},\{\mathbf{f}_i^{(0)}\}|\mathbf{A},v_i\in\mathcal{V}_1}(\mathbf{f}_i^{(1)}|v_i\in\mathcal{V}_1,\mathbf{A})$$

$$=\mathbb{E}_{\{y_i\},\{\mathbf{f}_i^{(0)}\}|\mathbf{A},v_i\in\mathcal{V}_1}\left(\frac{\mathbf{f}_i^{(0)}}{d_i+1}|v_i\in\mathcal{V}_1,\mathbf{A}\right) + \sum_{j\in\mathcal{N}_i}\mathbb{E}_{\{y_i\},\{\mathbf{f}_i^{(0)}\}|\mathbf{A},v_i\in\mathcal{V}_1}\left(\frac{\mathbf{f}_j^{(0)}}{\sqrt{d_i+1}\cdot\sqrt{d_j+1}}|v_i\in\mathcal{V}_1,\mathbf{A}\right)$$

$$=\frac{\boldsymbol{\mu}}{d_i+1} + \sum_{j\in\mathcal{N}_i}\left(\mathbb{E}_{\{y_i\},\{\mathbf{f}_i^{(0)}\}|\mathbf{A},v_i,v_j\in\mathcal{V}_1}\left(\frac{\mathbf{f}_j^{(0)}}{\sqrt{d_i+1}\cdot\sqrt{d_j+1}}|\mathbf{A},v_i,v_j\in\mathcal{V}_1\right)\cdot\mathbb{P}(v_j\in\mathcal{V}_1|\mathbf{A},v_i\in\mathcal{V}_1)\right.$$

$$\left.+\mathbb{E}_{\{y_i\},\{\mathbf{f}_i^{(0)}\}|\mathbf{A},v_i\in\mathcal{V}_1,v_j\in\mathcal{V}_2}\left(\frac{\mathbf{f}_j^{(0)}}{\sqrt{d_i+1}\cdot\sqrt{d_j+1}}|\mathbf{A},v_i\in\mathcal{V}_1,v_j\in\mathcal{V}_2\right)\cdot\mathbb{P}(v_j\in\mathcal{V}_2|\mathbf{A},v_i\in\mathcal{V}_1)\right) \tag{27}$$

Let $\mathbb{1}\{\cdot\}$ represent the indicator function: it equals 1 if and only if the event in the bracket holds.

$$\mathbb{E}_{\{y_i\},\{\mathbf{f}_i^{(0)}\}|\mathbf{A},v_i\in\mathcal{V}_1}\left(\sum_{j\in\mathcal{N}_i}\mathbb{1}\{v_j\in\mathcal{V}_1|\mathbf{A},v_i\in\mathcal{V}_1\}\right) = \sum_{j\in\mathcal{N}_i}\mathbb{P}(v_j\in\mathcal{V}_1|\mathbf{A},v_i\in\mathcal{V}_1) = d_i k_i. \tag{28}$$

Due to our assumptions in § 3, though the labels of nodes are dependent, they share the same marginal distribution and their joint distribution is a permutation-invariant function. Thus, given $v_i\in\mathcal{V}_1(y_i=1)$, the conditional joint distribution of $\{y_t|t\neq i\}$ is also permutation invariant—i.e., $\forall j,j'\in\mathcal{N}_i,\mathbb{P}(v_j\in\mathcal{V}_1|v_i\in\mathcal{V}_1)=\mathbb{P}(v_j'\in\mathcal{V}_1|v_i\in\mathcal{V}_1)$. Then for $\forall j\in\mathcal{N}_i$, we have: $d_i\mathbb{P}(v_j\in\mathcal{V}_1|v_i\in\mathcal{V}_1)=d_i k_i$ and $\mathbb{P}(v_j\in\mathcal{V}_1|v_i\in\mathcal{V}_1)=k_i$. Due to the independence of $\mathbf{A}$ and $\{y_i\}$, $\mathbb{P}(v_j\in\mathcal{V}_1|\mathbf{A},v_i\in\mathcal{V}_1)=k_i$.

$$\mathbb{E}_{\{y_i\},\{\mathbf{f}_i^{(0)}\}|v_i\in\mathcal{V}_1,\mathbf{A}}(\mathbf{f}_i^{(1)}|v_i\in\mathcal{V}_1,\mathbf{A}) = \frac{\boldsymbol{\mu}}{d_i+1} + \sum_{j\in\mathcal{N}_i}\left(\frac{k_i}{\sqrt{d_i+1}\cdot\sqrt{d_j+1}}\boldsymbol{\mu} - \frac{(1-k_i)}{\sqrt{d_i+1}\cdot\sqrt{d_j+1}}\rho\boldsymbol{\mu}\right)$$

$$=\frac{\boldsymbol{\mu}}{d_i+1} + \frac{((1+\rho)k_i-\rho)\boldsymbol{\mu}}{\sqrt{d_i+1}}\sum_{j\in\mathcal{N}_i}\frac{1}{\sqrt{d_j+1}}$$

$$=\left(\frac{1}{d_i+1} + \frac{\sum_{j\in\mathcal{N}_i}\frac{1}{\sqrt{d_j+1}}}{\sqrt{d_i+1}}((1+\rho)k_i-\rho)\right)\boldsymbol{\mu}$$

$$=\left(\frac{1}{d_i+1} + \frac{\sum_{j\in\mathcal{N}_i}\frac{\sqrt{d_i+1}}{\sqrt{d_j+1}}}{d_i+1}((1+\rho)k_i-\rho)\right)\boldsymbol{\mu}$$

$$=\left(\frac{1}{d_i+1} + \frac{d_i}{d_i+1}\frac{\sum_{j\in\mathcal{N}_i}r_{ij}}{d_i}((1+\rho)k_i-\rho)\right)\boldsymbol{\mu}. \tag{29}$$

Combining Equation (26) and Equation (29) and given the independence of $r_{ij}$ and $k_i$, we have:

$$\mathbb{E}_{\mathbf{A},\{y_i\},\{\mathbf{f}_i^{(0)}\}|d_i,v_i\in\mathcal{V}_1}(\mathbf{f}_i^{(1)}|v_i\in\mathcal{V}_1,d_i) = \frac{1}{d_i+1} + \frac{d_i}{d_i+1}\mathbb{E}_{\mathbf{A}|d_i,v_i\in\mathcal{V}_1}\left(\frac{\sum_{j\in\mathcal{N}_i}r_{ij}}{d_i}\right)\mathbb{E}_{\mathbf{A}|d_i,v_i\in\mathcal{V}_1}((1+\rho)k_i-\rho))\boldsymbol{\mu}$$

$$=\left(\frac{1+((1+\rho)h_i-\rho)d_i\overline{r_i}}{d_i+1}\right)\boldsymbol{\mu}\equiv\gamma_i^1\boldsymbol{\mu}. \tag{30}$$

Define $\epsilon \equiv (1 + \rho)h_i - \rho$, and we consider three cases: **(1)** $h_i \leq \frac{\rho}{1+\rho}$, **(2)** $h_i > \frac{\rho}{1+\rho}$ & $\overline{r_i} \leq \frac{1}{\epsilon}$, and **(3)** $h_i > \frac{\rho}{1+\rho}$ & $\overline{r_i} > \frac{1}{\epsilon}$.

- **CASE 1: $h_i \leq \frac{\rho}{1+\rho}$**

  • **Upper Bound**

  We have:
  $$\gamma_i^1 \leq \frac{1}{d_i + 1} \leq \frac{1}{2}. \tag{31}$$

  • **Lower Bound**

  To see if there exists a lower bound, we first show that when $h_i \leq \frac{\rho}{1+\rho}$, $\frac{1+((1+\rho)h_i-\rho)d_i\overline{r_i}}{d_i+1}$ is a decreasing function of $d_i$ given that $d_i \geq 1$.

  When $h_i \leq \frac{\rho}{1+\rho}$, we have:

  1. $((1 + \rho)h_i - \rho) \leq 0$
  2. $\frac{d_i}{d_i+1}$ is an increasing non-negative function of $d_i$
  3. $\overline{r_i}$ is an increasing non-negative function of $d_i$
  4. $\frac{1}{d_i+1}$ is a decreasing function of $d_i$

  Thus $\frac{1+((1+\rho)h_i-\rho)d_i\overline{r_i}}{d_i+1}$ is a decreasing function of $d_i$.

  When $h_i = \frac{\rho}{1+\rho}$, $\frac{1+((1+\rho)h_i-\rho)d_i\overline{r_i}}{d_i+1} = \frac{1}{d_i+1}$ and $0 < \frac{1}{d_i+1} \leq \frac{1}{2}$.

  When $h_i < \frac{\rho}{1+\rho}$,
  $$\begin{aligned} \frac{1 + ((1 + \rho)h_i - \rho)d_i\overline{r_i}}{d_i + 1} &\leq \frac{((1 + \rho)h_i - \rho)d_i\overline{r_i}}{d_i + 1} + \frac{1}{2} \\ &\leq \frac{((1 + \rho)h_i - \rho)\overline{r_i}}{2} + \frac{1}{2}. \end{aligned} \tag{32}$$

  And we know that when $h_i < \frac{\rho}{1+\rho}$:
  $$\lim_{\overline{r_i} \to \infty} \frac{((1 + \rho)h_i - \rho)\overline{r_i}}{2} + \frac{1}{2} = -\infty. \tag{33}$$

  Thus,
  $$\lim_{\overline{r_i} \to \infty} \frac{1 + ((1 + \rho)h_i - \rho)d_i\overline{r_i}}{d_i + 1} = -\infty. \tag{34}$$

- **CASE 2: $h_i > \frac{\rho}{1+\rho}$ & $\overline{r_i} \leq \frac{1}{\epsilon}$**

  If $h_i > \frac{\rho}{1+\rho}$, $0 < \epsilon \leq 1$; if $\overline{r_i} \leq \frac{1}{\epsilon}$, $0 < \epsilon\overline{r_i} \leq 1$. Given
  $$\mathbb{E}(\mathbf{f}_i^1) = \left( \frac{1 + \epsilon d_i\overline{r_i}}{d_i + 1} \right) \boldsymbol{\mu}. \tag{35}$$

  we have $0 < \frac{1}{d_i+1} < \gamma_i^1 \leq 1$.

- **CASE 3: $h_i > \frac{\rho}{1+\rho}$ & $\overline{r_i} > \frac{1}{\epsilon}$**

  In this case, Equation (35) still holds, since $h_i > \frac{\rho}{1+\rho}$.

  • **Lower Bound**

  If $\overline{r_i} > \frac{1}{\epsilon}$, then $\epsilon\overline{r_i} > 1$ and therefore $\gamma_i^1 > 1$.

- **Upper Bound**

When $\epsilon > 0$,

$$
\begin{aligned}
\frac{1 + \epsilon d_i \overline{r_i}}{d_i + 1} &> \frac{\epsilon d_i \overline{r_i}}{d_i + 1} \\
&\geq \frac{\epsilon \overline{r_i}}{2}.
\end{aligned}
\tag{36}
$$

Because

$$
\lim_{\overline{r_i} \to \infty} \frac{\epsilon \overline{r_i}}{2} = \infty,
\tag{37}
$$

we have:

$$
\lim_{\overline{r_i} \to \infty} \frac{1 + \epsilon d_i \overline{r_i}}{d_i + 1} = \infty.
\tag{38}
$$

To sum it up,

$$
\gamma_i^1 \in
\begin{cases}
(-\infty, \frac{1}{2}], & \text{if } h_i \leq \frac{\rho}{1+\rho} \\
(0, 1], & \text{if } h_i > \frac{\rho}{1+\rho} \ \& \ \overline{r_i} \leq \frac{1}{(1+\rho)h_i - \rho} \\
(1, \infty), & \text{otherwise.}
\end{cases}
\tag{39}
$$

**Remarks: Special cases**

For different types of graphs and different nodes in the graph, $d_i \overline{r_i}$ might be different. For a **d-regular graph** whose nodes have constant degree or a graph whose adjacency matrix is **row-normalized**, we will have:

$$
\begin{aligned}
\gamma_i^1 &= \left( \frac{1 + ((1+\rho)h_i - \rho)d_i \overline{r_i}}{d_i + 1} \right) \\
&\leq \left( \frac{1}{(d_i + 1)} + \frac{d_i}{(d_i + 1)} \right) = 1
\end{aligned}
\tag{40}
$$

Equality is achieved if and only if $h_i = 1$. To note that, $h_i = 1$ is not achievable for every node as long as there are more than one class. **Boundary nodes** will suffer most.

$\square$

## A.4 Proof of Theorem 2

*Proof.* After propagating over multiple layers, we use $\xi_i^l$ to account for the accumulated scaling effects ($\gamma_i^l$). That is, $\mathbb{E}_{\mathbf{A}, \{y_i\}, \{\mathbf{f}_i^{(0)}\}|d_i} \left( \frac{\mathbf{f}_i^{(l)}}{d_i + 1} | d_i \right) = \xi_i^l \boldsymbol{\mu}$. Furthermore, conditioned on $\mathbf{A}$ and $\xi_i^l$, the set of nodes contributing positively to node $v_i$ is defined as:

$\hat{\mathcal{N}}_i^s(\mathbf{A}, \xi_i^l) \equiv \{v_j | v_j \in \mathcal{N}_i \text{ and } (\xi_i^l \boldsymbol{\mu}) \cdot \mathbb{E}_{\{y_i\}, \{\mathbf{f}_i^{(0)}\}|\xi_i^l, \mathbf{A}} \left( {\mathbf{f}_j^{(l)}}^T | \xi_i^l, \mathbf{A} \right) > 0\}.$

$\mathbb{E}_{\mathbf{A}, \{y_i\}, \{\mathbf{f}_i^{(0)}\}|d_i, \xi_i^l, v_j \in \mathcal{N}_i} \left( \mathbf{f}_j^{(l)} r_{ij} | d_i, \xi_i^l, v_j \in \mathcal{N}_i \right) =
\begin{cases}
\xi_i^{l'} \boldsymbol{\mu} & v_j \in \hat{\mathcal{N}}_i^s(\mathbf{A}, \xi_i^l) \\
-\rho_i^l \xi_i^{l'} \boldsymbol{\mu} & v_j \notin \hat{\mathcal{N}}_i^s(\mathbf{A}, \xi_i^l)
\end{cases}$

In the following proof, we use $\hat{\mathcal{N}}_i^s$ to refer to $\hat{\mathcal{N}}_i^s(\mathbf{A}, \xi_i^l)$ for conciseness.

The representations at $(l+1)$-th layer:

$$
\mathbf{f}_i^{(l+1)} = \frac{\mathbf{f}_i^{(l)}}{d_i + 1} + \sum_{j \in \mathcal{N}_i} \frac{\mathbf{f}_i^{(l)}}{\sqrt{d_i + 1} \cdot \sqrt{d_j + 1}}
\tag{41}
$$

$$\mathbb{E}_{\mathbf{A},\{y_i\},\{\mathbf{f}_i^{(0)}\}|d_i,\xi_i^l} \left( \mathbf{f}_i^{(l+1)}|d_i,\xi_i^l \right) \tag{42}$$

$$= \mathbb{E}_{\mathbf{A},\{y_i\},\{\mathbf{f}_i^{(0)}\}|d_i,\xi_i^l} \left( \frac{\mathbf{f}_i^{(l)}}{d_i+1}|d_i,\xi_i^l \right) + \mathbb{E}_{\mathbf{A},\{y_i\},\{\mathbf{f}_i^{(0)}\}|d_i,\xi_i^l} \left( \sum_{j\in\mathcal{N}_i} \frac{\mathbf{f}_j^{(l)}}{\sqrt{d_i+1}\cdot\sqrt{d_j+1}}|d_i,\xi_i^l \right) \tag{43}$$

$$= \frac{\xi_i^l\boldsymbol{\mu}}{d_i+1} + \frac{1}{d_i+1}\mathbb{E}_{\mathbf{A},\{y_i\},\{\mathbf{f}_i^{(0)}\}|d_i,\xi_i^l} \left( \sum_{j\in\mathcal{N}_i} \frac{\mathbf{f}_j^{(l)}\sqrt{d_i+1}}{\sqrt{d_j+1}}|d_i,\xi_i^l \right) \tag{44}$$

In § 3, we assume that $\{d_j\}$ follow the same distribution and their joint distribution function is permutation-invariant, and $\{y_j\}$ and $\{\mathbf{f}_j^l\}$ also have this property. Thus, conditioned on $d_i$, $\xi_i^l$ and $\mathcal{N}_i$, the distribution of $\frac{\mathbf{f}_j^{(l)}\sqrt{d_i+1}}{\sqrt{d_j+1}}$ is the same for $\forall j \in \mathcal{N}_i$ (neighbors are indistinguishable) and we obtain:

$$\mathbb{E}_{\mathbf{A},\{y_i\},\{\mathbf{f}_i^{(0)}\}|d_i,\xi_i^l,\mathcal{N}_i} \left( \sum_{j\in\mathcal{N}_i} \frac{\mathbf{f}_j^{(l)}\sqrt{d_i+1}}{\sqrt{d_j+1}}|d_i,\xi_i^l,\mathcal{N}_i \right) \tag{45}$$

$$= d_i\mathbb{E}_{\mathbf{A},\{y_i\},\{\mathbf{f}_i^{(0)}\}|d_i,\xi_i^l,\mathcal{N}_i,v_j\in\mathcal{N}_i} \left( \frac{\mathbf{f}_j^{(l)}\sqrt{d_i+1}}{\sqrt{d_j+1}}|d_i,\xi_i^l,\mathcal{N}_i,v_j\in\mathcal{N}_i \right) \tag{46}$$

Given that the joint distribution of $\{d_j\}$, the joint distribution of $\{y_j\}$ and the joint distribution of $\{\mathbf{f}_j^l\}$ are permutation-invariant functions, for any $\mathcal{N}_i$, $\mathbb{E}_{\mathbf{A},\{y_i\},\{\mathbf{f}_i^{(0)}\}|d_i,\xi_i^l,\mathcal{N}_i} \left( \sum_{j\in\mathcal{N}_i} \frac{\mathbf{f}_j^{(l)}\sqrt{d_i+1}}{\sqrt{d_j+1}}|d_i,\xi_i^l,\mathcal{N}_i \right)$ is the same. That is:

$$\mathbb{E}_{\mathbf{A},\{y_i\},\{\mathbf{f}_i^{(0)}\}|d_i,\xi_i^l} \left( \sum_{j\in\mathcal{N}_i} \frac{\mathbf{f}_j^{(l)}\sqrt{d_i+1}}{\sqrt{d_j+1}}|d_i,\xi_i^l \right) \tag{47}$$

$$= \mathbb{E}_{\mathbf{A},\{y_i\},\{\mathbf{f}_i^{(0)}\}|d_i,\xi_i^l,\mathcal{N}_i} \left( \sum_{j\in\mathcal{N}_i} \frac{\mathbf{f}_j^{(l)}\sqrt{d_i+1}}{\sqrt{d_j+1}}|d_i,\xi_i^l,\mathcal{N}_i \right) \tag{48}$$

$$= d_i\mathbb{E}_{\mathbf{A},\{y_i\},\{\mathbf{f}_i^{(0)}\}|d_i,\xi_i^l,v_j\in\mathcal{N}_i} \left( \frac{\mathbf{f}_j^{(l)}\sqrt{d_i+1}}{\sqrt{d_j+1}}|d_i,\xi_i^l,v_j\in\mathcal{N}_i \right) \tag{49}$$

We note that $v_j$ in Equation 49 can be any node except $v_i$ due to the equivalence of those nodes.

Combining Equation 44 and Equation 49, we can obtain:

$$\mathbb{E}_{\mathbf{A},\{y_i\},\{\mathbf{f}_i^{(0)}\}|d_i,\xi_i^l}\left(\mathbf{f}_i^{(l+1)}|d_i,\xi_i^l\right) \tag{50}$$

$$= \frac{\xi_i^l\boldsymbol{\mu}}{d_i+1} + \frac{d_i}{d_i+1}\mathbb{E}_{\mathbf{A},\{y_i\},\{\mathbf{f}_i^{(0)}\}|d_i,\xi_i^l,v_j\in\mathcal{N}_i}\left(\frac{\mathbf{f}_j^{(l)}\sqrt{d_i+1}}{\sqrt{d_j+1}}|d_i,\xi_i^l,v_j\in\mathcal{N}_i\right) \tag{51}$$

$$= \frac{\xi_i^l\boldsymbol{\mu}}{d_i+1} + \frac{d_i}{d_i+1}\left(\mathbb{E}_{\mathbf{A},\{y_i\},\{\mathbf{f}_i^{(0)}\}|d_i,\xi_i^l,v_j\in\hat{\mathcal{N}}_i^s,v_j\in\mathcal{N}_i}\left(\frac{\mathbf{f}_j^{(l)}\sqrt{d_i+1}}{\sqrt{d_j+1}}|d_i,\xi_i^l,v_j\in\hat{\mathcal{N}}_i^s,v_j\in\mathcal{N}_i\right) \right. \tag{52}$$

$$\left. \cdot \ \mathbb{P}\left(v_j\in\hat{\mathcal{N}}_i^s|v_j\in\mathcal{N}_i,d_i,\xi_i^l\right) + \mathbb{E}_{\mathbf{A},\{y_i\},\{\mathbf{f}_i^{(0)}\}|d_i,\xi_i^l,v_j\notin\hat{\mathcal{N}}_i^s,v_i\in\mathcal{N}_i}\left(\frac{\mathbf{f}_j^{(l)}\sqrt{d_i+1}}{\sqrt{d_j+1}}|d_i,\xi_i^l,v_j\notin\hat{\mathcal{N}}_i^s,v_j\in\mathcal{N}_i\right) \right. \tag{53}$$

$$\left. \cdot \ \mathbb{P}\left(v_j\notin\hat{\mathcal{N}}_i^s,|d_i,\xi_i^l,v_j\in\mathcal{N}_i\right)\right) \tag{54}$$

$$= \frac{\xi_i^l\boldsymbol{\mu}}{d_i+1} + \left(\frac{d_i\mathbb{P}(v_j\in\hat{\mathcal{N}}_i^s|d_i,\xi_i^l,v_j\in\mathcal{N}_i)}{d_i+1}\xi_i^{l'}\boldsymbol{\mu} - \frac{d_i(1-\mathbb{P}(v_j\in\hat{\mathcal{N}}_i^s|d_i,\xi_i^l,v_j\in\mathcal{N}_i))}{d_i+1}\rho_i^l\xi_i^{l'}\boldsymbol{\mu}\right) \tag{55}$$

$$= \frac{\xi_i^l\boldsymbol{\mu}}{d_i+1} + \frac{d_i(\mathbb{P}(v_j\in\hat{\mathcal{N}}_i^s|d_i,\xi_i^l,v_j\in\mathcal{N}_i)(1+\rho_i^l)-\rho_i^l)\xi_i^{l'}\boldsymbol{\mu}}{d_i+1} \tag{56}$$

Define $\bar{r}_i^l \equiv \frac{\xi_i^{l'}}{\xi_i^l}$. If we know the degree $d_i$, $\xi_i^l$, and that a neighbor $v_j$ is in the group $\hat{\mathcal{N}}_i^s$, $\bar{r}_i^l$ actually represents the ratio of expected $\mathbf{f}_j^l r_{i,j}$ to expected $\mathbf{f}_i^l$. We regard it as the effective related degree at $l$-th layer. For the initial layer, if we know the degree $d_i$, $y_i$, and that a neighbor $v_j$ is in the group $\mathcal{N}_i^s$, the ratio of $\mathbf{f}_j^0 r_{i,j}$ to $\mathbf{f}_i^0$ is $r_{ij}$. Recall that the related degree $\overline{r_i}$ at the initial layer is the expected average of $r_{ij}$ in the neighborhood. Thus $\bar{r}_i^l$ is an extension of $\overline{r_i}$. Moreover, let $\hat{h}_i^l = \mathbb{P}(v_j\in\hat{\mathcal{N}}_i^s|v_j\in\mathcal{N}_i,d_i,\xi_i^l)$ represent the probability of a neighbor whose representation has a positive contribution in expectation. This naturally extends the meaning of homophily in deeper layers. Thus, we regard it as the effective homophily of node $v_i$ at $l$-th layer. Given the degree $d_i$, $\xi_i^l$, $\rho_i^l$ represents the ratio of the probability that a neighbor will have a positive rather than a negative contribution to $v_i$. This naturally extends the meaning of $\rho$.

We further write Equation 56 as:

$$\frac{\xi_i^l}{d_i+1}\left(1+\left((\hat{h}_i^l(1+\rho_i^l)-\rho_i^l)d_i\bar{r}_i^l\right)\right)\boldsymbol{\mu}$$

and it will have three cases similar to Thm. 1.

$\square$

## A.5 PROOF OF THEOREM 3

*Proof.* Similar to Equation 26, we can express the conditional expectation of $\mathbf{f}_i^{(1)}$ as:

$$
\begin{aligned}
&\mathbb{E}_{\mathbf{A},\{y_i\},\{\mathbf{f}_i^{(0)}\}|d_i,v_i\in\mathcal{V}_1}\left(\mathbf{f}_i^{(1)}|v_i\in\mathcal{V}_1,d_i\right)\\
=&\mathbb{E}_{m_i^0|v_i\in\mathcal{V}_1,d_i}\left(\mathbb{E}_{\mathbf{A},\{y_i\},\{\mathbf{f}_i^{(0)}\}|d_i,v_i\in\mathcal{V}_1,m_i^0}\left(\mathbf{f}_i^{(1)}|v_i\in\mathcal{V}_1,d_i,m_i^0\right)\right)\\
=&\mathbb{E}_{m_i^0|v_i\in\mathcal{V}_1,d_i}\left(\mathbb{E}_{\mathbf{A},\{y_i\},\{\mathbf{f}_i^{(0)}\}|v_i\in\mathcal{V}_1,d_i,m_i^0}\left(\frac{\mathbf{f}_i^{(0)}}{d_i+1}|v_i\in\mathcal{V}_1,d_i,m_i^0\right)\right.\\
&+\mathbb{E}_{\mathbf{A},\{y_i\},\{\mathbf{f}_i^{(0)}\}|v_i\in\mathcal{V}_1,d_i,m_i^0}\left(\sum_{j\in\mathcal{N}_i}\frac{\mathbf{f}_j^{(0)}}{\sqrt{d_i+1}\cdot\sqrt{d_j+1}}|v_i\in\mathcal{V}_1,d_i,m_i^0\right)\right)\\
=&\frac{\boldsymbol{\mu}}{d_i+1}\\
&+\mathbb{E}_{m_i^0|v_i\in\mathcal{V}_1,d_i}\left(\mathbb{E}_{\mathbf{A}|d_i,v_i\in\mathcal{V}_1,m_i^0}\left(\sum_{j\in\mathcal{N}_i}\left(\mathbb{E}_{\{y_i\},\{\mathbf{f}_i^{(0)}\}|\mathbf{A},v_i,v_j\in\mathcal{V}_1,m_i^0}\left(\frac{\mathbf{f}_j^{(0)}}{\sqrt{d_i+1}\cdot\sqrt{d_j+1}}|\mathbf{A},m_i^0,v_j\in\mathcal{V}_1,v_i\in\mathcal{V}_1\right)\right.\right.\right.\\
&\cdot\mathbb{P}(v_j\in\mathcal{V}_1|m_i^0,v_i\in\mathcal{V}_1,\mathbf{A})\\
&\left.\left.\left.+\mathbb{E}_{\{y_i\},\{\mathbf{f}_i^{(0)}\}|\mathbf{A},v_i\in\mathcal{V}_1,v_j\in\mathcal{V}_2,m_i^0}\left(\frac{\mathbf{f}_j^{(0)}}{\sqrt{d_i+1}\cdot\sqrt{d_j+1}}|\mathbf{A},m_i^0,v_i\in\mathcal{V}_1,v_j\in\mathcal{V}_2\right)\cdot\mathbb{P}(v_j\in\mathcal{V}_2|m_i^0)\right)\right)\right).
\end{aligned}
\tag{57}
$$

Next, we will show how to compute the conditional expectation and conditional probability in the summand.

$$
\begin{aligned}
&\mathbb{E}_{\{y_i\},\{\mathbf{f}_i^{(0)}\}|\mathbf{A},v_i,v_j\in\mathcal{V}_1,m_i^0}\left(\frac{\mathbf{f}_j^{(0)}}{\sqrt{d_i+1}\cdot\sqrt{d_j+1}}|m_i^0,v_i,v_j\in\mathcal{V}_1,\mathbf{A}\right)\\
=&\mathbb{E}_{\{y_i\},\{\mathbf{f}_i^{(0)}\}|\mathbf{A},v_i\in\mathcal{V}_1,m_i^0,v_j}\left(\frac{\mathbf{f}_j^{(0)}}{\sqrt{d_i+1}\cdot\sqrt{d_j+1}}|\mathbf{A},m_i^0,v_i\in\mathcal{V}_1,v_j\in\mathcal{V}_1,v_j\text{wrongly send information}\right)\\
&\cdot\mathbb{P}(v_j\text{wrongly send information}|\mathbf{A},m_i^0,v_i\in\mathcal{V}_1,v_j\in\mathcal{V}_1)\\
&+\mathbb{E}_{\{y_i\},\{\mathbf{f}_i^{(0)}\}|\mathbf{A},v_i\in\mathcal{V}_1,m_i^0,v_j}\left(\frac{\mathbf{f}_j^{(0)}}{\sqrt{d_i+1}\cdot\sqrt{d_j+1}}|\mathbf{A},m_i^0,v_i\in\mathcal{V}_1,v_j\in\mathcal{V}_1,v_j\text{correctly send information}\right)\\
&\cdot\mathbb{P}(v_j\text{correctly send information}|\mathbf{A},m_i^0,v_i\in\mathcal{V}_1,v_j\in\mathcal{V}_1).
\end{aligned}
\tag{58}
$$

Combined with the independence assumption, we have:

$$
\mathbb{P}(v_j\text{wrongly send information}|\mathbf{A},m_i^0,v_i\in\mathcal{V}_1,v_j\in\mathcal{V}_1)=m_i^0.
\tag{59}
$$

Similarly, we obtain:

$$
\mathbb{P}(v_j\text{correctly send information}|\mathbf{A},m_i^0,v_i\in\mathcal{V}_1,v_j\in\mathcal{V}_1)=1-m_i^0.
\tag{60}
$$

Then Equation (58) can be rewritten as:

$$
\mathbb{E}_{\{y_i\},\{\mathbf{f}_i^{(0)}\}|\mathbf{A},v_i,v_j\in\mathcal{V}_1,m_i^0}\left(\frac{\mathbf{f}_j^{(0)}}{\sqrt{d_i+1}\cdot\sqrt{d_j+1}}|m_i^0,v_i,v_j\in\mathcal{V}_1,\mathbf{A}\right)
$$

$$
=\mathbb{E}_{\{y_i\},\{\mathbf{f}_i^{(0)}\}|\mathbf{A},v_i\in\mathcal{V}_1,m_i^0,v_j}\left(\frac{\mathbf{f}_j^{(0)}}{\sqrt{d_i+1}\cdot\sqrt{d_j+1}}|\mathbf{A},m_i^0,v_i\in\mathcal{V}_1,v_j\in\mathcal{V}_1,v_j\text{wrongly send information}\right)\cdot m_i^0
$$

$$
+\mathbb{E}_{\{y_i\},\{\mathbf{f}_i^{(0)}\}|\mathbf{A},v_i\in\mathcal{V}_1,m_i^0,v_j}\left(\frac{\mathbf{f}_j^{(0)}}{\sqrt{d_i+1}\cdot\sqrt{d_j+1}}|\mathbf{A},m_i^0,v_i\in\mathcal{V}_1,v_j\in\mathcal{V}_1,v_j\text{correctly send information}\right)\cdot(1-m_i^0)
$$

$$
=-\frac{m_i^0\boldsymbol{\mu}}{\sqrt{d_i+1}\cdot\sqrt{d_j+1}}+\frac{(1-m_i^0)\boldsymbol{\mu}}{\sqrt{d_i+1}\cdot\sqrt{d_j+1}}
$$

$$
=\frac{(1-2m_i^0)\boldsymbol{\mu}}{\sqrt{d_i+1}\cdot\sqrt{d_j+1}}.
$$

$$(61)$$

Similarly, we have:
$$
\mathbb{E}_{\{y_i\},\{\mathbf{f}_i^{(0)}\}|\mathbf{A},v_i\in\mathcal{V}_1,v_j\in\mathcal{V}_2,m_i^0}\left(\frac{\mathbf{f}_j^{(0)}}{\sqrt{d_i+1}\cdot\sqrt{d_j+1}}|\mathbf{A},m_i^0,v_i\in\mathcal{V}_1,v_j\in\mathcal{V}_2\right)=\frac{(1-2m_i^0)\rho\boldsymbol{\mu}}{\sqrt{d_i+1}.\cdot\sqrt{d_j+1}}.
$$

$$(62)$$

Consider the independence between $m_i^0$ and node class & degrees, and insert Equation (61) and Equation (62) into Equation (57), we will have:

$$
\mathbb{E}_{\mathbf{A},\{y_i\},\{\mathbf{f}_i^{(0)}\}|d_i,v_i\in\mathcal{V}_1}\left(\mathbf{f}_i^{(1)}|v_i\in\mathcal{V}_1,d_i\right)
$$

$$
=\frac{\boldsymbol{\mu}}{d_i+1}+\mathbb{E}_{m_i^0|v_i\in\mathcal{V}_1,d_i}\left(\mathbb{E}_{\mathbf{A}|d_i,v_i\in\mathcal{V}_1,m_i^0}\left(\sum_{j\in\mathcal{N}_i}(\frac{(1-2m_i^0)k_i\boldsymbol{\mu}}{\sqrt{d_i+1}\cdot\sqrt{d_j+1}}+\frac{(1-2m_i^0)(1-k_i)\rho\boldsymbol{\mu}}{\sqrt{d_i+1}\cdot\sqrt{d_j+1}})\right)\right)
$$

$$
=\frac{\boldsymbol{\mu}}{d_i+1}+\mathbb{E}_{m_i^0|v_i\in\mathcal{V}_1,d_i}\left(\mathbb{E}_{\mathbf{A}|d_i,v_i\in\mathcal{V}_1,m_i^0}\left(\sum_{j\in\mathcal{N}_i}\frac{(1-2m_i^0)(\rho+(1-\rho)k_i)\boldsymbol{\mu}}{\sqrt{d_i+1}\cdot\sqrt{d_j+1}}\right)\right)
$$

$$
=\frac{\boldsymbol{\mu}}{d_i+1}+\mathbb{E}_{m_i^0|v_i\in\mathcal{V}_1,d_i}\left(\left(\frac{(1-2m_i^0)(\rho+(1-\rho)h_i)d_i\overline{r_i}}{d_i+1}\right)\boldsymbol{\mu}\right)
$$

$$
=\left(\frac{1+(1-2e_i^0)(\rho+(1-\rho)h_i)d_i\overline{r_i}}{d_i+1}\right)\boldsymbol{\mu}\equiv\gamma_i^1\mathbb{E}(\mathbf{f}_i^{(0)}).
$$

$$(63)$$

When $h_i\leq 1$, $\rho+(1-\rho)h_i=(1-h_i)\rho+h_i>0$. Define $\epsilon'\equiv(1-2e_i^0)(\rho+(1-\rho)h_i)$. To obtain the ranges of $\gamma_i^1$, when $e_i^0\geq 0.5$, $\epsilon'\leq 0$, it resembles the derivations of CASE 1 in Proof A.3; when $e_i^0<0.5$, $0<\epsilon'\overline{r_i}\leq 1$, it resembles the derivations of CASE 2; when $e_i^0<0.5$, $\epsilon'\overline{r_i}>1$, it resembles the derivations of CASE 3. □

# B ADDITIONAL EXPERIMENTS

## B.1 ABLATION STUDY

Table B.1: Ablation study: degree correction has consistent benefits (robust to oversmoothing & ability to handle heterophily) in different datasets while signed information has more benefits in heterophilous datasets. Best performance of each model is highlighted in gray.

| Layers | 2 | 4 | 8 | 16 | 32 | 64 | 2 | 4 | 8 | 16 | 32 | 64 |
|---|---|---|---|---|---|---|---|---|---|---|---|---|
| | **Cora** ($h$=0.81) | | | | | | **Citeseer** ($h$=0.74) | | | | | |
| Base | $86.56_{\pm1.21}$ | $86.04_{\pm0.72}$ | $85.51_{\pm1.51}$ | $85.33_{\pm0.72}$ | $85.37_{\pm1.58}$ | $72.17_{\pm8.89}$ | $76.51_{\pm1.63}$ | $75.03_{\pm1.67}$ | $73.96_{\pm1.52}$ | $73.59_{\pm1.51}$ | $71.91_{\pm1.94}$ | $32.08_{\pm15.74}$ |
| +deg | $86.72_{\pm1.29}$ | $86.02_{\pm0.97}$ | $85.49_{\pm1.32}$ | $85.27_{\pm1.59}$ | $85.27_{\pm1.51}$ | $84.21_{\pm1.22}$ | $76.63_{\pm1.38}$ | $74.64_{\pm1.97}$ | $74.15_{\pm1.61}$ | $73.73_{\pm1.31}$ | $73.61_{\pm1.84}$ | $70.56_{\pm2.27}$ |
| +sign | $84.81_{\pm1.63}$ | $86.06_{\pm1.7}$ | $85.67_{\pm1.26}$ | $85.39_{\pm0.97}$ | $84.85_{\pm0.98}$ | $78.57_{\pm6.73}$ | $77.13_{\pm1.69}$ | $74.56_{\pm2.02}$ | $73.64_{\pm1.65}$ | $72.31_{\pm2.32}$ | $71.98_{\pm3.44}$ | $68.68_{\pm6.72}$ |
| +deg,sign | $86.96_{\pm1.38}$ | $86.20_{\pm0.89}$ | $85.63_{\pm0.78}$ | $85.47_{\pm1.18}$ | $85.55_{\pm1.66}$ | $77.81_{\pm7.95}$ | $76.81_{\pm1.71}$ | $74.68_{\pm1.97}$ | $74.69_{\pm2.35}$ | $73.28_{\pm1.45}$ | $71.81_{\pm2.28}$ | $69.91_{\pm3.97}$ |
| | **Cornell** ($h$=0.3) | | | | | | **Chameleon** ($h$=0.23) | | | | | |
| Base | $61.89_{\pm3.72}$ | $60.00_{\pm5.24}$ | $58.92_{\pm5.24}$ | $56.49_{\pm5.73}$ | $58.92_{\pm3.15}$ | $49.19_{\pm16.70}$ | $64.98_{\pm1.84}$ | $62.65_{\pm3.09}$ | $62.43_{\pm3.28}$ | $54.69_{\pm2.58}$ | $47.68_{\pm2.63}$ | $29.74_{\pm5.21}$ |
| +deg | $63.78_{\pm5.57}$ | $62.70_{\pm5.90}$ | $59.46_{\pm4.52}$ | $56.49_{\pm5.73}$ | $57.57_{\pm4.20}$ | $58.92_{\pm3.15}$ | $66.54_{\pm2.19}$ | $68.31_{\pm2.70}$ | $68.99_{\pm2.38}$ | $67.68_{\pm3.70}$ | $56.86_{\pm8.80}$ | $41.95_{\pm9.56}$ |
| +sign | $85.41_{\pm7.27}$ | $76.76_{\pm7.07}$ | $70.00_{\pm5.19}$ | $67.57_{\pm9.44}$ | $63.24_{\pm6.07}$ | $63.24_{\pm6.53}$ | $65.31_{\pm3.20}$ | $53.55_{\pm6.35}$ | $53.05_{\pm2.28}$ | $51.93_{\pm4.00}$ | $57.17_{\pm3.39}$ | $51.93_{\pm8.95}$ |
| +deg,sign | $84.32_{\pm6.37}$ | $78.92_{\pm8.09}$ | $73.51_{\pm5.90}$ | $70.81_{\pm5.64}$ | $68.11_{\pm5.14}$ | $62.43_{\pm6.67}$ | $65.75_{\pm1.81}$ | $61.49_{\pm7.38}$ | $53.73_{\pm7.79}$ | $52.43_{\pm5.37}$ | $55.92_{\pm5.14}$ | $56.95_{\pm3.93}$ |

We now study the impact of two proposed mechanisms (signed messages and degree correction, § 4). To better show their effects, we add each design choice to a base model and track the changes in the node classification performance. As the base model, we choose a GCN variant that uses the message passing mechanism (Kipf & Welling, 2016) with weight bias, a residual connection (but *not decaying aggregation*) for robustness to oversmoothing, and Elu non-linearity $\sigma$. We denote the model that replaces the message passing with our signed messages mechanism as `+sign`, and the model that incorporates the degree correction as `+deg`. The model that uses both designs is denoted as `+sign,deg`. Table B.1 gives the accuracy of the models in the supervised node classification task for different layers.

We observe that both mechanisms alleviate the oversmoothing problem. Specifically, the base model has a sharp performance decrease after 32 layers, while the other models have significantly higher performance. In general, the `+deg` model is better than `+sign` in alleviating oversmoothing, and has a consistent performance gain across different data. For Chameleon, we observe increase in accuracy as we stack more layers; the large performance gain of GGCN results from the degree correction. In Cora and Cornell, `+sign,deg` consistently performs better across different layers than the model with only one design, demonstrating the constructive effect from our two proposed mechanisms.

The signed message design has an advantage in heterophilous datasets. In the Cornell dataset, using signed information rather than plain message passing results in over 20% gain, which explains the strong performance of GGCN. However, in homophilous datasets, the benefit from signed messages is limited, as these datasets have few different-class neighbors. This is because when the effective homophily $\widehat{h}_i^l$ is high and error rate $e_i^0$ is low, $\lim_{\widehat{h}_i^l \to 1, e_i^0 \to 0} \frac{(1+\rho)\widehat{h}_i^l - \rho}{(1-2e_i^0)(\rho+(1-\rho)\widehat{h}_i^l)} = 1$, so using signed messages is less beneficial in homophilous datasets.

## B.2 BATCH NORM & LAYER NORM

In § 4, we use decaying aggregation instead of other normalizing mechanisms. Other mechanisms, such as batch or layer norm, may be seen as solutions to the heterophily and oversmoothing problems. However, batch norm cannot compensate for the dispersion of the mean vectors (§ 3) due to different degrees and homophily levels of the nodes. Although, to some extent, it reduces the speed by which the representations of the susceptible nodes (case 1 & 2) move towards the other class (good for oversmoothing), it also prevents the representations of the nodes that could benefit from the propagation (case 3) from increasing the distances (drop in accuracy). Layer norm is better at overcoming the dispersion effect but may lead to a significant accuracy drop in some datsets when a subset of features are more important than the others. Thus, we **do not** use any of these normalizations.

Next, we provide experiments to show the effects of batch norm and layer norm. We use the following base model (the same model used in §B.1): a GCN Kipf & Welling (2016) with weight bias, Elu non-linearity and residual connection. We do **not** include any of our designs so as to exclude any other factors that can affect the performance. The models we compare against are `+BN` and `+LN`, which represent the models that add batch norm and layer norm right before the non-linear activation, respectively.

Table B.2: Effects of using batch norm & layer norm: decrease in accuracy but improvement in oversmoothing. Best performance of each model across different layers is highlighted in gray.

| Layers | 2 | 4 | 8 | 16 | 32 | 64 | 2 | 4 | 8 | 16 | 32 | 64 |
|---|---|---|---|---|---|---|---|---|---|---|---|---|
| | **Cora** (h=0.81) | | | | | | **Citeseer** (h=0.74) | | | | | |
| Base | 86.56±1.21 | 86.04±0.72 | 85.51±1.51 | 85.33±0.72 | 85.37±1.58 | 72.17±8.89 | 76.51±1.63 | 75.03±1.67 | 73.96±1.52 | 73.59±1.51 | 71.91±1.94 | 32.08±15.74 |
| +BN | 84.73±1.10 | 83.76±1.61 | 83.94±1.51 | 84.57±1.22 | 84.63±1.58 | 85.17±1.18 | 71.62±1.48 | 71.58±1.00 | 72.18±1.39 | 72.45±1.42 | 72.76±1.31 | 72.61±1.41 |
| +LN | 84.73±1.63 | 86.60±1.01 | 86.72±1.36 | 86.08±1.16 | 85.67±1.23 | 85.13±1.20 | 76.11±1.80 | 74.02±2.77 | 75.00±1.95 | 74.50±0.96 | 74.49±2.10 | 73.94±2.03 |
| | **Cornell** (h=0.3) | | | | | | **Chameleon** (h=0.23) | | | | | |
| Base | 61.89±3.72 | 60.00±5.24 | 58.92±5.24 | 56.49±5.73 | 58.92±3.15 | 49.19±16.70 | 64.98±1.84 | 62.65±3.09 | 62.43±3.28 | 54.69±2.58 | 47.68±2.63 | 29.74±5.21 |
| +BN | 58.38±6.42 | 59.19±4.59 | 55.41±6.65 | 57.30±3.15 | 57.57±6.29 | 57.02±6.19 | 60.88±2.24 | 61.38±2.17 | 61.84±4.08 | 61.97±3.01 | 59.04±3.79 | 57.84±3.67 |
| +LN | 58.11±6.19 | 55.68±6.19 | 58.92±7.63 | 59.19±3.07 | 58.92±3.15 | 58.00±3.03 | 61.86±1.73 | 62.17±2.48 | 62.41±2.99 | 60.37±2.36 | 58.25±3.03 | 58.92±3.15 |

Table B.2 shows that both batch norm and layer norm can help with oversmoothing. Moreover, adding layer norm is in general better than adding batch norm. This is expected because the scaling effect caused by the propagation can be alleviated by normalizing across the node representations. Thus, the dispersion of the expected representations can be mitigated. On the other hand, batch norm normalizes across all the nodes, so it requires sacrificing the nodes that benefit (case 3) to compensate

for the nodes that are prone to moving towards the other classes (case 1 & case 2). As a result, batch norm is less effective in mitigating oversmoothing and leads to a bigger decrease in accuracy.

Another finding is that both layer norm and batch norm lead to a significant accuracy decrease (2%-3%) in the heterophilous datasets. +BN has a clear accuracy drop even in the homophilous datasets. As Thm. 1 points out: higher heterophily level may result in sign flip. If the representations flip the sign, using batch norm or layer norm will not revert the sign, but they may instead encourage the representations to move towards the other class more.

Given the limitations shown above, we **do not** use either batch norm or layer norm in our proposed model, GGCN.

### B.3 MORE ON THE INITIAL & DEVELOPING STAGES

Section 5.4 shows how the node classification accuracy changes for nodes of different degrees with the number of layers on Citeseer. Here, we provide more details of this experiment and give the results on another dataset, Cora.

**Datasets.** According to Thm. 1 and 2, in order to see both the initial and the developing stage, we need to use homophilous datasets. In heterophilous datasets, most nodes satisfy case 1, so the initial stage does not exist.

**Measurement of effective homophily $\widehat{h}_i^l$.** To estimate the effective homophily $\widehat{h}_i^l$, we measure the portion of neighbors that have the same ground truth label as $v_i$ and are correctly classified. Following our theory, we estimate $\widehat{h}_i^l$ **before** the last propagation takes place and analyze its impact on the accuracy of the final layer. In more detail, we obtain the node representations before the last propagation from a trained vanilla GCN Kipf & Welling (2016) and then perform a linear transformation using the weight matrix and bias vector from the last layer. Then, we use the transformed representations to classify the neighbors of node $v_i$ and compute $\widehat{h}_i^l$. We note that we leverage the intermediate representations only for the estimation of $\widehat{h}_i^l$; the accuracy of the final layer is still measured using the outputs from the final layer.

**Degree intervals.** To investigate the change in GCN accuracy with different layers for nodes with different degrees, we categorize the nodes in $n$ degree intervals. For the degree intervals, we use logarithmic binning (base 2). In detail, we denote the highest and lowest degree by $d_{max}$ and $d_{min}$, respectively, and let $\Omega \equiv \frac{\log_2 d_{max} - \log_2 d_{min}}{n}$. Then, we divide the nodes into $n$ intervals, where the $j$-th interval is defined as: $[d_{min} \cdot 2^{(j-1)\Omega}, d_{min} \cdot 2^{j\Omega})$.

**Dataset: Citeseer.** Figure 1 and Table B.3 show how the accuracy changes with the number of layers for different node degree intervals. We observe that in the initial stage, the accuracy increases as the degree and $\widehat{h}_i^l$ increase. However, in the developing stage, the accuracy of high-degree nodes drops more sharply than that of low-degree nodes.

Table B.3: Citeseer: Accuracy (Acc) and average effective homophily ($\widehat{h}_i^l$) for nodes with different degrees across various layers. Last layer of initial stage marked in gray.

| Layers | | Degrees | | | | |
|---|---|---|---|---|---|---|
| | | $[1, 2]$ | $[3, 6]$ | $[7, 15]$ | $[16, 39]$ | $[40, 99]$ |
| 2 | Acc | 74.44 | 78.51 | 84.04 | 96.00 | 100.00 |
| | $\widehat{h}_i^l$ | 0.65 | 0.67 | 0.72 | 0.83 | 0.91 |
| 3 | Acc | 70.92 | 77.59 | 83.03 | 92.75 | 100.00 |
| | $\widehat{h}_i^l$ | 0.63 | 0.66 | 0.71 | 0.84 | 0.92 |
| 4 | Acc | 60.54 | 68.56 | 77.63 | 94.00 | 100.00 |
| | $\widehat{h}_i^l$ | 0.54 | 0.58 | 0.66 | 0.79 | 0.84 |
| 5 | Acc | 41.66 | 48.70 | 56.97 | 61.33 | 11.11 |
| | $\widehat{h}_i^l$ | 0.36 | 0.38 | 0.45 | 0.48 | 0.11 |
| 6 | Acc | 23.61 | 28.87 | 41.17 | 31.83 | 0.00 |
| | $\widehat{h}_i^l$ | 0.18 | 0.19 | 0.27 | 0.22 | 0.00 |

**Dataset: Cora.** The results for Cora are shown in Figure B.1 and Table B.4. In the initial stage, the nodes with lower

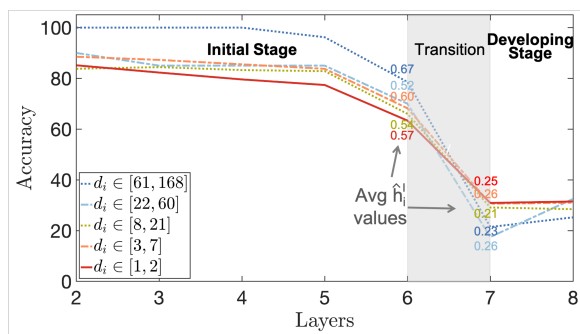

Figure B.1: Cora: Accuracy per (logarithmic) degree bin.

Table B.4: Cora: Accuracy and average effective homophily $(\widehat{h}_i^l)$ for nodes with different degrees across different layers. Last layer of initial stage marked in gray.

| Layers | | Degrees | | | | |
|---|---|---|---|---|---|---|
| | | $[1,2]$ | $[3,7]$ | $[8,21]$ | $[22,60]$ | $[61,168]$ |
| 2 | Acc | 85.14 | 88.52 | 83.78 | 90.00 | 100.00 |
| | $\widehat{h}_i^l$ | 0.77 | 0.77 | 0.72 | 0.63 | 0.81 |
| 3 | Acc | 82.28 | 87.26 | 84.42 | 85.00 | 100.00 |
| | $\widehat{h}_i^l$ | 0.77 | 0.78 | 0.73 | 0.64 | 0.84 |
| 4 | Acc | 79.57 | 85.52 | 83.26 | 85.00 | 100.00 |
| | $\widehat{h}_i^l$ | 0.75 | 0.76 | 0.71 | 0.63 | 0.80 |
| 5 | Acc | 77.38 | 83.75 | 82.83 | 85.00 | 96.19 |
| | $\widehat{h}_i^l$ | 0.72 | 0.73 | 0.68 | 0.56 | 0.83 |
| 6 | Acc | 63.35 | 68.20 | 66.01 | 70.00 | 78.57 |
| | $\widehat{h}_i^l$ | 0.57 | 0.60 | 0.54 | 0.52 | 0.67 |
| 7 | Acc | 30.91 | 30.62 | 29.12 | 17.50 | 21.43 |
| | $\widehat{h}_i^l$ | 0.25 | 0.26 | 0.21 | 0.26 | 0.23 |
| 8 | Acc | 31.48 | 31.02 | 28.44 | 32.50 | 25.24 |
| | $\widehat{h}_i^l$ | 0.24 | 0.26 | 0.20 | 0.28 | 0.24 |

degrees usually have lower accuracy. One exception is the nodes with degrees in the range $[3,7]$. These nodes have higher accuracy because the average effective homophily $\widehat{h}_i^l$ of that degree group is the second highest. In the developing stage, the accuracy of the high-degree nodes drops more than the accuracy of the remaining node groups.

GCN's behavior on both Citesser and Cora datasets verifies our conjecture based on our theorems in § 3.3.

## B.4 HOMOPHILY METRIC

In Table 1, we provide the graph homophily level $h$, which is defined as the average of the nodes' homophily levels. In our theoretical analysis in § 3, we use a node's homophily level and relative degree to characterize three cases (Fig. 2 and Thm. 1). To better align our empirical results with our theoretical analysis, we provide below the proportion of nodes in each case.

Table B.5: The percentage (%) of nodes in each case (Fig. 2 and Thm. 1): case 1: $h_i \leq \frac{\rho}{1+\rho}$, case 2: $h_i > \frac{\rho}{1+\rho}$ & $\overline{r_i} \leq \frac{1}{(1+\rho)h_i - \rho}$ and case 3: otherwise. The dominant case for each dataset is colored in grey.

| | Texas | Wisconsin | Actor | Squirrel | Chameleon | Cornell | Citeseer | Pubmed | Cora |
|---|---|---|---|---|---|---|---|---|---|
| Hom. level $h$ | 0.11 | 0.21 | 0.22 | 0.22 | 0.23 | 0.3 | 0.74 | 0.8 | 0.81 |
| #Nodes | 183 | 251 | 7,600 | 5,201 | 2,277 | 183 | 3,327 | 19,717 | 2,708 |
| #Edges | 295 | 466 | 26,752 | 198,493 | 31,421 | 280 | 4,676 | 44,327 | 5,278 |
| #Classes | 5 | 5 | 5 | 5 | 5 | 5 | 7 | 3 | 6 |
| Case 1 | 87.43 | 78.49 | 63.49 | 47.74 | 48.48 | 79.23 | 18.33 | 14.80 | 6.50 |
| Case 2 | 8.74 | 15.93 | 29.53 | 50.14 | 48.84 | 6.01 | 25.58 | 33.75 | 32.98 |
| Case 3 | 3.83 | 5.58 | 6.99 | 2.11 | 2.68 | 14.75 | 56.09 | 51.45 | 60.52 |

We observe that a low graph homophily level indicates that case 1 is the dominant case, while a high graph homophily level indicates that case 3 is the dominant case. As we stated in Thm 1, under "non-swapping" condition, case 3 is the only case to have a benefit. Thus, in the commonly-used homophilous datasets (Citeseer, Pubmed and Cora), case 3 dominates and enjoys the benefits from graph convolution. In the Texas, Wisconsin and Actor datasets, only case 1 dominates and case 1 hurts the performance most, this explains why GCN performs worse than most methods in Table 1. In the Squirrel and Chameleon datasets, both case 1 and case 2 dominate, GCN yields acceptable results. Using portions of the three cases to analyze the datasets is a better metric, because it aligns more with the GCN's performance. As shown by Table B.5, Wisconsin, Actor, Squirrel and Chameleon have very close graph level homophily, but GCN's performance are quite different as shown in Table 1. Our way to define homophily level does predict that GCN should perform decently in Squirrel and Chameleon datasets.

### B.5 COMPLEXITY ANALYSIS

We first analyze the time complexity of the forward path GGCN. For degree correction, we need to compute the $r_{ij}$. However, we do not need to compute all of them, and we only need to compute $r_{ij}$ for node pairs who are linked. The time complexity is $O(|\mathcal{E}|)$, but it is a one-time computation, the results of which can be saved. We compute $\tau_{ij}^l$ based on the learned weights $\lambda_0^l, \lambda_1^l$. The time complexity is $O(|\mathcal{E}| \cdot L)$, where $L$ is the number of layers. The time complexity to compute the signed function is $O(|\mathcal{E}| \cdot H \cdot L)$, where $H$ is the hidden dimension of representations, and we only compute the cosine similarity between the nodes that are linked. We also need to compute the multiplication of the propagation matrix and representation matrices. The time complexity is $O(|\mathcal{V}|^2 \cdot H \cdot L)$, . Similarly, the time complexity of the multiplication of representation matrices and the weight matrices is $O(|\mathcal{V}| \cdot H^2 \cdot L)$. The total time complexity would be $O(\max(|\mathcal{V}|^2, |\mathcal{E}|) \cdot H \cdot L)$. The complexity of GGCN resembles the complexity of attention-based model. Thus we compare with GAT (Veličković et al., 2017) the total time to train and test on 4 larger homophilous and heterophilous datasets. We run the training and testing for 10 times.

Table B.6: Training and test time (m:minutes, s:seconds) for 10 runs. Shorter time is colored in grey.

|  | **Actor** | **Chameleon** | **Citeseer** | **Cora** |
|---|---|---|---|---|
| **#Nodes** | 7,600 | 2,277 | 3,327 | 2,708 |
| **#Edges** | 26,752 | 31,421 | 4,676 | 5,278 |
| GGCN | $7m24s$ | $2m29s$ | $2m9s$ | $5m47s$ |
| GAT | $8m14s$ | $7m34s$ | $9m39s$ | $4m11s$ |

In general, GGCN runs faster than GAT because degree correction and signed messages in GGCN learn fewer parameters.

## C HYPERPARAMETERS AND PARAMETERS

### C.1 HYPERPARAMETER SETTINGS

**Experiments for Table 1 & Table 2**

For the baselines, we set the same hyperparameters that are provided by the original papers or the authors' github repositories, and we match the results they reported in their respective papers. In our experiments, we find that the original hyperparameters set by the authors are already well-tuned.

All the models use Adam as the optimizer. GAT sets the initial learning rate as 0.005 and Geom-GCN uses a custom learning scheduler. All the other models (include GGCN) use the initial learning rate 0.01.

For GGCN, we use the following hyperparameters:

- $k$ in the decaying aggregation: 3

We tune the parameters in the following ranges:

- Dropout rate: [0.0, 0.7]
- Weight decay: [1e-7, 1e-2]
- Hidden units: {8, 16, 32, 64, 80}
- Decay rate $\eta$: [0.0, 1.5]

**Experiments for Table B.1 and Table B.2**

The hyperparameters that are used in all the models (Base, `+deg`, `+sign`, `+deg,sign`, `+BN`, `+LN`) are set to be the same and they are tuned for every dataset. Those common hyperparameters are:

- Dropout rate: [0.0, 0.7]
- Weight decay: [1e-7, 1e-2]
- Hidden units: {8, 16, 32, 64, 80}

## C.2 INITIALIZATION OF PARAMETERS

**Initialization**

For GGCN, we adopt the following parameter initialization in the experiments for Table 1 & Table 2

- Initialization of $\lambda_0^l$ and $\lambda_1^l$: 0.5 and 0, respectively
- Initialization of $\alpha^l$, $\beta_0^l$, $\beta_1^l$ and $\beta_2^l$: 2, 0, 0, 0, respectively.

We initialize $\beta_{\{0,1,2\}}^l = 0$ in Eq. (6) because, after applying softmax, $\hat{\beta}_{\{0,1,2\}}^l = 1/3$; this ensures equal contributions from positive and negative neighbors and themselves (and sum=1). We initialize $\lambda_1^l = 0$ in Eq. (9) following the common practice for initializing the bias. We set $\alpha^l = 2$ and $\lambda_0^l = 0.5$ in Eq. (6) and Eq. (9), because when $r_{ij} \to +\infty$, the degree correction (including global scaling) is: $\hat{\alpha}^l \tau_{ij}^l = \text{softplus}(2) \cdot \text{softplus}(0.5 \cdot (0-1) + 0) \approx 1$. As we mention in Thm. 1, when the homophily level is high, nodes with large $\overline{r_i}$ may benefit (case 3), thus we do not want to compensate for these nodes and would like to keep $\hat{\alpha}^l \tau_{ij}^l$ close to 1.

## C.3 PARAMETERS AFTER TRAINING

Figure C.1 shows the original and corrected $\overline{r_{ij}}$ (avg $r_{ij}$). Corrected $r_{ij}$ are given by $\hat{\alpha}^l \tau_{ij}^l r_{ij}$, which are a combination of global scaling and local degree correction. Because $\beta_{\{0,1,2\}}^l$ sum to 1 and do not change global scaling, they are not considered in the corrected $r_{ij}$. As can be seen in Fig. C.1, after the training, GGCN learns to increase $r_{ij}$, which satisfies our theorems.

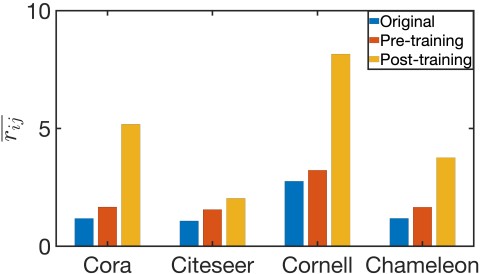

Figure C.1: Original $\overline{r_{ij}}$ (avg. $r_{ij}$) and corrected (pre- and post-training).

## D TABLE 2 WITH LARGER FONTS

Table D.1: Model performance for different layers: mean accuracy ± stdev over different data splits. Per dataset and GNN model, we also report the layer at which the best performance (given in Table 1) is achieved. 'OOM': out of memory; 'INS': numerical instability.

**Cora (h=0.81)**

| Layers | 2 | 4 | 8 | 16 | 32 | 64 | Best |
|---|---|---|---|---|---|---|---|
| GGCN (ours) | 87.00±1.15 | 87.48±1.32 | 87.63±1.33 | 87.51±1.19 | 87.95±1.05 | 87.28±1.41 | 32 |
| GPRGNN | 87.93±1.11 | 87.95±1.18 | 87.87±1.41 | 87.26±1.51 | 87.18±1.29 | 87.32±1.21 | 4 |
| H2GCN* | 87.87±1.20 | 86.10±1.51 | 86.18±2.10 | OOM | OOM | OOM | 2 |
| GCNII* | 85.35±1.56 | 85.35±1.48 | 86.38±0.98 | 87.12±1.11 | 87.95±1.23 | 88.37±1.25 | 64 |
| PairNorm | 85.79±1.01 | 85.07±0.91 | 84.65±1.09 | 82.21±2.84 | 60.32±8.28 | 44.39±5.60 | 2 |
| Geom-GCN* | 85.35±1.57 | 21.01±2.61 | 13.98±1.48 | 13.98±1.48 | 13.98±1.48 | 13.98±1.48 | 2 |
| GCN | 86.98±1.27 | 83.24±1.56 | 31.03±3.08 | 31.05±2.36 | 30.76±3.43 | 31.89±2.08 | 2 |
| GAT | 87.30±1.10 | 86.50±1.20 | 84.97±1.24 | INS | INS | INS | 2 |

**Cornell (h=0.3)**

| Layers | 2 | 4 | 8 | 16 | 32 | 64 | Best |
|---|---|---|---|---|---|---|---|
| GGCN (ours) | 83.78±6.73 | 83.78±6.16 | 84.86±5.69 | 83.78±6.73 | 83.78±6.51 | 84.32±5.90 | 6 |
| GPRGNN | 76.76±8.22 | 77.57±7.46 | 80.27±8.11 | 78.38±6.04 | 74.59±7.66 | 70.00±5.73 | 8 |
| H2GCN* | 81.89±5.98 | 82.70±6.27 | 80.27±6.63 | OOM | OOM | OOM | 1 |
| GCNII* | 67.57±11.34 | 64.59±9.63 | 73.24±5.91 | 77.84±3.97 | 75.41±5.47 | 73.78±4.37 | 16 |
| PairNorm | 50.27±7.17 | 53.51±8.00 | 58.38±5.01 | 58.38±3.01 | 58.92±3.15 | 58.92±3.15 | 32 |
| Geom-GCN* | 60.54±3.67 | 23.78±11.64 | 12.97±2.91 | 12.97±2.91 | 12.97±2.91 | 12.97±2.91 | 2 |
| GCN | 60.54±5.30 | 59.19±3.30 | 58.92±3.15 | 58.92±3.15 | 58.92±3.15 | 58.92±3.15 | 2 |
| GAT | 61.89±5.05 | 58.38±4.05 | 58.38±3.86 | INS | INS | INS | 2 |

**Citeseer (h=0.74)**

| Layers | 2 | 4 | 8 | 16 | 32 | 64 | Best |
|---|---|---|---|---|---|---|---|
| GGCN (ours) | 76.83±1.82 | 76.77±1.48 | 76.91±1.56 | 76.88±1.56 | 76.97±1.52 | 76.65±1.38 | 10 |
| GPRGNN | 77.13±1.67 | 77.05±1.43 | 77.09±1.62 | 76.00±1.64 | 74.97±1.47 | 74.41±1.65 | 2 |
| H2GCN* | 76.90±1.80 | 76.09±1.54 | 74.10±1.83 | OOM | OOM | OOM | 1 |
| GCNII* | 75.42±1.78 | 75.29±1.90 | 76.00±1.66 | 76.96±1.38 | 77.33±1.48 | 77.18±1.47 | 32 |
| PairNorm | 73.59±1.47 | 72.62±1.97 | 72.32±1.58 | 59.71±15.97 | 27.21±10.95 | 23.82±6.64 | 2 |
| Geom-GCN* | 78.02±1.15 | 23.01±1.95 | 7.23±0.87 | 7.23±0.87 | 7.23±0.87 | 7.23±0.87 | 2 |
| GCN | 76.50±1.36 | 64.33±8.27 | 24.18±1.71 | 23.07±2.95 | 25.3±1.77 | 24.73±1.66 | 2 |
| GAT | 76.55±1.23 | 75.33±1.39 | 66.57±5.08 | INS | INS | INS | 2 |

**Chameleon (h=0.23)**

| Layers | 2 | 4 | 8 | 16 | 32 | 64 | Best |
|---|---|---|---|---|---|---|---|
| GGCN (ours) | 70.77±1.42 | 69.58±2.68 | 70.33±1.70 | 70.44±1.82 | 70.29±1.62 | 70.20±1.95 | 5 |
| GPRGNN | 46.58±1.771 | 45.72±3.45 | 41.16±5.79 | 39.58±7.85 | 35.42±8.52 | 36.38±2.40 | 2 |
| H2GCN* | 59.06±1.85 | 60.11±2.15 | OOM | OOM | OOM | OOM | 4 |
| GCNII* | 61.07±4.10 | 63.86±3.04 | 62.89±1.18 | 60.20±2.10 | 56.97±1.81 | 55.99±2.27 | 4 |
| PairNorm | 62.74±2.82 | 59.01±2.80 | 54.12±2.24 | 46.38±2.23 | 46.78±2.26 | 46.27±3.24 | 2 |
| Geom-GCN* | 60.00±2.81 | 19.17±1.66 | 19.58±1.73 | 19.58±1.73 | 19.58±1.73 | 19.58±1.73 | 2 |
| GCN | 64.82±2.24 | 53.11±4.44 | 35.15±3.14 | 35.39±3.23 | 35.20±3.25 | 35.50±3.08 | 2 |
| GAT | 60.26±2.50 | 48.71±2.96 | 35.09±3.55 | INS | INS | INS | 2 |

