# OpenReview forum: "Two Sides of the Same Coin: Heterophily and Oversmoothing in Graph Convolutional Neural Networks"
_ICLR.cc/2022/Conference — ICLR 2022 Submitted_

### Official Review · Reviewer_1EPP · 2021-10-30

**Correctness:** 2
**Technical Novelty And Significance:** 3
**Empirical Novelty And Significance:** 3
**Recommendation:** 5
**Confidence:** 5

**Main Review:**

Strength:
- The idea of finding the common causes of the two problems instead of simply releasing them separately is very enlightening. It is a good starting point for tackling these two problems from a theoretical perspective.
- Theoretical analysis is provided and it is technically sound.
- This paper justifies the introduction of the signed messages in solving these two problems in a more general setting.

Weakness:
- Theorems are based on probably many assumptions.
- The theorems fail to prove that the relative degree would affect the misclassification rate. On one hand, in theorem 1, high-homophily nodes with a relatively low degree will move toward the original boundary and the opposite class in the shallow layers. However, based on this theorem, they will not flip the labels because the parameters will never be negative. On the other hand, the effective homophily of the nodes is actually the only factor in theorem 2. The proof of Theorem 2 in appendix A.4 actually verifies that the relative degree has no effect in this process.
- In this paper, high/low homophily/heterophily are defined by comparing the node's homophily level with the homophily level of the whole graph. When we discuss the heterophily problem faced by the GCNs, we are mostly talking about the heterophily level of the graph. Unluckily, the homophily level defined in this paper cannot be translated into the homophily level of the graph. That is to say, a graph with an average homophily level higher than 0.5 doesn't guarantee that it has more nodes with a homophily level higher than the average. At the same time, the empirical experiments are conducted on a collective level (performance on the whole graph on average) instead of on each node. I have to say the theory and the experiment are not well combined.


**Summary Of The Paper:**

Two issues confronting GCNs, heterophily and over-smoothing, share a common cause. This article asserts that the relative degree and homophily level account for node representation movement and further affect the likelihood of misclassification.

The contributions of this paper are as follows:
1) it attempts to establish a connection between heterophily and over-smoothing by identifying common causes;
2) it establishes that the relative degree and homophily level have an effect on the misclassification rate; and
3) it also designs and presents a model that allows for negative interactions between nodes, which is shown to be robust to over-smoothing and capable of achieving state-of-the-art performance on heterophily datasets.

**Summary Of The Review:**

Overall, I recommend rejecting this paper. It is actually exciting to see a paper that tries to link the two widely existing problems of GCNs together theoretically. However, although the theorems presented in the paper are actually correct, they don't support the conclusion that the authors claim.

After reading the other reviews, I would choose to maintain my opinion, and the lack of innovation is my primary concern.

---

> ### Author Response · Authors · 2021-11-22
> **Response to Reviewer 1EPP**
>
> #### ****The relation of relative degree and misclassification rate****
>
> In the ***general response*** (***relation between theorems and conclusions*** section), we have illustrated the high-level logic of our theorems: Lemma 2 proves that moving closer to the original decision boundary leads to a non-decreasing misclassification rate; in Theorem 1 we show that high-homophily nodes with a relatively low degree will move toward the original boundary and the opposite class in the shallow layers. Combining the conclusions from the theories, we show that a small relative degree will have a negative impact on the misclassification rate: low-degree nodes with high homophily move closer to the decision boundary (Thm 1), and non-zero movement towards the decision boundary leads to non-decreasing misclassification (Lemma 2). ****We note that $\gamma_i^1$ does not need to be negative to cause an increase in the misclassification rate.**** $\{\mathbf{f}_i^{(l)}\}$ are random vectors and may have some probability of being misclassified. In the proof of Lemma 2, we compute the probability of being misclassified through the integration of PDF.
>
> #### ****Too many assumptions****
>
> Thanks for your comments. We have updated our assumptions so that they are more general. We do not assume specific graph generation processes. We also updated our proofs in the appendix (especially Thm 2) to accommodate that. Those changes do not change our final formulas and conclusions.
>
> We agree that we make several assumptions to make this problem tractable. As a starting point for exploring the relation between heterophily and oversmoothing problems, the simplified theoretical models do provide us with meaningful insights and useful practices. We showed empirically that those insights and practices are effective in real scenarios: model with non-linearity and multi-class setting. Though theories for more general settings may be preferable, the non-linear nature of neural networks makes the full analysis prohibitively hard. Thus, in this work, we adopt the linear assumption, which is very common in the literature (e.g., [Zhu, NeurIPS 2020; Oono, ICLR 2019; Li, AAAI 2018]) and the binary classification assumption as in [Baranwa, ICML 2021], and show the applicability of our theoretical results to more general settings in our empirical analysis. Furthermore, most existing analysis for oversmoothing adopts the spectral perspective (they neglect non-linearity and weights), which makes it hard to understand how the performance gradually degenerates and how different kinds of nodes (with different homophily levels and degrees) behave. Our work is a complement to those works.
>
> #### ****Relative degree has no effect in Theorem 2****
> Actually it has effect in Theorem 2. Since we have updated our assumptions and proofs, we explain it based on the new version. In the new version, page 5 (the blue text), the factor that appear in the formula is the effective relative degree of $l$-th layer, which is defined as $\bar{r_i}^{l} \equiv \frac{{\xi_{i}^{l}}'}{{\xi_{i}^{l}}}$. As we will show later, $\bar{r_i}^{l}$ closely relates to $\bar{r_i}$.
>
> Intuitively speaking,  when the representation of an arbitrary neighbor $v\_j$ is positively correlated to the representation of  $v\_i$,  $\xi\_{i}^{l}$ is the expected scaling of $\mathbf{f}\_i^{l+1}$,
> and ${\xi\_{i}^{l}}'$
> is the expected scaling of $\mathbf{f}\_{j} \cdot r_{ij}$,  which are defined as follows:
> $\mathbb{E}(\mathbf{f}\_i^{l+1})=\xi\_{i}^{l}\mathbf{\mu}$ and
> $\mathbb{E}(\mathbf{f}\_{j} \cdot r\_{ij})={\xi\_{i}^{l}}'\mathbf{\mu}$.
> At the initial layer, when the $v\_i$ and $v\_j$ are in the same class, the ratio of  $\mathbf{f}\_{j}^{0}r\_{i,j}$ to $\mathbf{f}\_{i}^{0}$ is $r_{ij}$ and $\bar{r_i}$ is the expected average of $r_{ij}$. Thus, $\bar{r_i}^{l}$
> is a natural extension of $\bar{r_i}$ (expected $r_{ij}$).
> ***For a more rigorous definition and reasoning, please check the main paper, page 3 for current updated assumptions, page 5 (theorem 2, the blue text) for math definitions and appendix, page 17 (blue text) for explanations.***
>
>
> #### ****Global and local homophily****
> We agree that using global homophily larger or smaller than 0.5 is a coarse way to distinguish homophilous and heterophilous graphs, but as we discuss next it serves as a good proxy for more subtle and computationally expensive measures. A better way is to compute the portion of each of the three cases in Fig. 2, and decide based on the dominant case. We provide the ratio per case and dataset in ***appendix B.4***. We find that the characterization of nodes using node's homophily level and relative degrees are a better indicator to predict the GCN's performance. We have a detailed discussion in ***appendix B.4***. Though the absolute global homophily level is not sufficient to fully characterize how heterophilous the graphs are, it serves as a fast-to-compute measure to indicate the dominant case.

---

> > ### Author Response · Authors · 2021-12-02
> > **Regarding our response**
> >
> > Dear reviewer:
> >
> > Thanks for taking the time to review our paper and response. For the technical novelty, we think our main contribution is to reveal the relationship between the two problems, which is not discovered/mentioned in prior works. For prior works in oversmoothing, the majority of the papers analyze the problem from the spectral perspective, which cannot be used to reveal the relationship to heterophily. Our way to analyze it is orthogonal and a complement to the current analysis. As side information, our paper is finished in early Feb (we have evidence for that) and some works (that focus on model designs) are actually concurrent works by that time, but still, we have added them and compared with them later.
> >
> > By taking into your suggestions, we have added additional experiments, relaxed the assumptions, and modified the proofs. For additional experiments, we actually verify that our analysis can provide a more accurate and better prediction of the performance of GCN than the global homophily measure.
> > For the current assumptions, we do not require independence of the node degrees, do not require independence of node labels/features, and we do not assume a specific way to generate the graphs. The modified assumptions satisfy the property of real graphs. We also make efforts in updating the proofs to accommodate the relaxation of the assumptions.
> >
> > Regarding your initial questions about the relationship between the theory and the conclusions, we have also provided a detailed explanation to clear your confusion.
> >
> > After reviewing the rebuttal, if you think we have cleared most of your initial questions, would you kindly consider raising your score to give credits to the efforts we have made in the rebuttal?

---

### Official Review · Reviewer_ehjS · 2021-11-01

**Correctness:** 3
**Technical Novelty And Significance:** 3
**Empirical Novelty And Significance:** Not applicable
**Recommendation:** 6
**Confidence:** 3

**Main Review:**

(+)

- I really like the study of the authors, the approach in terms of degree and homophily level, the analysis in terms of separability is interesting. And the suggested mechanisms seem to be useful.

- The appendix is detailed and provides a lot of additional information and insights.

(-)

- The paper is there to present the work and should be self-contained. However, below the theorems, it just says "proof in appendix". Even if the authors do not think it is necessary that the readers get an idea why the proposals are valid, an intuition should be given, at least in the appendix.

- The authors mention the related works from Bo et al. and Chien et al. which also apply signed messages. Although the theoretical setting is slightly different in this paper, the novelty of the proposed solution is limited.

- A closely related recent work is not mentioned/discussed in the paper:
Lim et al. Large Scale Learning on Non-Homophilous Graphs: New Benchmarks and Strong Simple Methods, NeurIPS 2021

(other)
Oversmoothing is particularly a problem in dense graphs, e.g., DDI data. I wonder if the authors have any insights (from other experiments or similar) that the theory about the nodes with varying degree holds also for such data sets.


**Summary Of The Paper:**

The paper analyses the GNN over-smoothing problem in terms of node degree and homophily level, suggests two correction mechanisms, and empirically shows that they help in various experiments.


**Summary Of The Review:**

This approach to study this topic definitely deserves the attention of the community. The paper is overall well-written. I do not recommend full acceptance at the current point because I think it is not fair scientific practice to omit recent and very closely related work altogether.

[Note] I reviewed the paper before. While the previous version of the paper was basically the same in the technical and experimental contents, the presentation was lacking clarity a lot. Great parts have been rewritten now, and I appreciate the efforts of the authors in addressing these issues. Since there was already a lot of discussion of various aspects of the approach, I only have few comments above. I am fairly confident about the paper itself, but I did not verify the proofs in the appendix in more detail.

---

> ### Author Response · Authors · 2021-11-22
> **Response to Reviewer ehjS**
>
> #### ****Novelty and contribution****
> In the ***general response***(it contains the information you are interested in), we highlighted that our main contribution is the analysis of the relation between the heterophily and oversmoothing problems, and not the introduction of new designs. The designs are used to verify the findings and insights from our theoretical analysis. That said, our proposed designs have some key differences from those in prior works, as we use different design choices and contribute new knowledge (under previously unexplored settings).
>
> To the best of our knowledge, [Tang et al., CIKM 2020] is the only prior work that utilizes different degree information to improve the node classification performance. However, the design is used for homophilous graphs and it is _not_ shown that the degree information can help with oversmoothing. On the other hand, our degree design is inspired by and is closely related to our theory, and is used to address both problems; thus, the way we incorporate degree information is different from prior work.
>
> [Bo et al., AAAI 2021] and [Chien et al., ICLR 2021] allow signed aggregation of graph filters, which is motivated differently from ours (they aim to design a more universal graph filter) and thus we design differently. Moreover, neither of these prior works shows how signed messages can help with the oversmoothing problem and both of them neglect the fact that signed messages can sometimes be harmful. Thus, our work provides additional insights and useful practices.
>
> #### ****Intuition of the proof****
> Due to the limit in the main text, we have added the intuition of the proof in appendix A (especially blue text in Lemma 2, proof for theorem 1 and theorem 2).
>
> #### ****Related work****
> Thank you for these pointers. We have added the related paper to our revised paper.
>
> #### ****Other****
> Dense graphs-> easy homophily?
> Here we provide our reasoning using our theorems. The relative degree $\bar{r_i}$ will be small in dense graphs (the extreme case is a fully connected graph where $\bar{r_i}=1$ for every node). In that situation, case 2 (Fig. 2) is the dominant case or case 2 constitutes a significant portion. As we said, case 3 is the only case that will benefit from graph convolution. More portion of case 2 will be more vulnerable to oversmoothing. In ***appendix B.4***, we provide the percentages of cases that appear in each graph. For the common homophily dataset we use, like Cora, Citeseer and Pubmed, case 3 is the dominant case. Among the three datasets, Pubmed is denser than the other two, and you can see a drop in the percentage of case 3 and an increase in the percentage of case 2.

---

> > ### Comment · Reviewer_ehjS · 2021-11-30
> > **Thank you!**
> >
> > I think the discussion with the other reviewers has shown that it would be good if the presentation got improved, the main point of a paper is to communicate findings after all. I must say that I do not have concrete recommendations though and admit that it's a complex work. Overall, I therefore think that my rating is fair.

---

> > > ### Author Response · Authors · 2021-11-30
> > > **Thanks for your feedback**
> > >
> > > Thanks for your feedback and suggestions!

---

### Official Review · Reviewer_nAQJ · 2021-11-03

**Correctness:** 3
**Technical Novelty And Significance:** 3
**Empirical Novelty And Significance:** 2
**Recommendation:** 6
**Confidence:** 3

**Main Review:**

In this paper, the authors attempt to theoretically unify the cause of oversmoothing and heterophily. Specifically, the authors prove that under their proposed framework with certain assumptions, the level of homophily and relative degree may affect model performance. The authors further show the link between GNN performance and signed messages. Based on the analysis results, the authors propose a GCN model named GGCN by utilizing signed messages and degree corrections along with decaying aggregation. The effectiveness of the proposed model is validated by several experiments using real-world datasets of different statistics. In general, the paper is structured well and easy to follow. However, there are several issues the authors need to address.

First, it would be better if the authors can further justify the novelty of the proposed approach, especially the modeling part. Both signed messages and degree corrections have been adopted to improve GNN performance and to solve a similar or the same problem (e.g., heterophily). I suggest the authors to further justify why this approach can be considered novel rather than combining several existing methods that focus on the same problem.

I also have two questions for the evaluation: 1) It is good to see that GGCN performs well on heterophilous datasets. However, I suggest the authors to further discuss the potential reasons why GGCN performs worse than several baselines given that the analysis in Sec. 3 does not show the signed messages and degree corrections may hurt model performance on homophilous graphs. 2) The ablation study results in Table B.1 seem to be not aligned with the results in Table 2. When the number of layer increases, GGCN's performance seems to be stable in Table 1 but all the variants seem to decay much in Table B.1. I suggest the authors to further explain the difference between these results.

The signed messages require to compute pair-wise cosine similarities on node embeddings. It would be better if the authors can append a scalability analysis to compare GGCN with baselines on training/inference efficiency.

**Summary Of The Paper:**

In this paper, the authors attempt to theoretically unify the cause of oversmoothing and heterophily, propose a generalized GCN model named GGCN by using signed messages and degree corrections along with decaying aggregation, and conduct several experiments to validate the performance of the proposed method.

**Summary Of The Review:**

Strength:
+ Comprehensive summary on existing literature
+ Good paper structure

Weakness:
- Novelty needs to be justified
- Evaluation results need to be further discussed
- Missing scalability analysis

---

> ### Author Response · Authors · 2021-11-22
> **Response to Reviewer nAQJ**
>
> #### **Novelty and contribution**
> In the ***general response***(it contains information you are interested in), we highlighted that our main contribution is the analysis of the relation between the heterophily and oversmoothing problems, and not the introduction of new designs. The designs are used to verify the findings and insights from our theoretical analysis. That said, our proposed designs have some key differences from those in prior works and contribute new knowledge in previously unexplored settings.
>
> To the best of our knowledge, [Tang et al., CIKM 2020] is the only prior work that utilizes degree information to improve the node classification performance. However, the design is used for homophilous graphs and it is not shown that the degree information can help with oversmoothing. On the other hand, our degree design is inspired by and is closely related to our theory, and is used to address both problems; thus, the way we incorporate degree information is different from prior work.
>
> [Bo et al., AAAI 2021] and [Chien et al., ICLR 2021] allow signed aggregation of graph filters, which is motivated differently from ours (they aim to design a more universal graph filter) and thus we design differently. Moreover, neither of these prior works shows how signed messages can help with the oversmoothing problem and both of them neglect the fact that signed messages can sometimes be harmful. Thus, our work provides additional insights and useful practices.
>
> #### ****Difference between Table 2 and Table B.1****
> In Table 2, we use the GGCN model which incorporates 3 strategies into the GCN model. On the other hand, in Table B.1 we use different models: We chose the base model as GCN+residual connection and then added our proposed designs on top of this base model. The reason is that GCN+residual is a simple but effective model against oversmoothing compared to pure GCN as shown by [Rong, ICLR 2020], so using it as the base model is more convincing than using GCN. As for the performance decay in Table B.1, I would say that the two strategies are useful in alleviating oversmoothing, but it does not mean it can obtain stable results when added to every base model. Also, strategy 3 is also helpful, especially when combined with our two strategies.
>
> [Rong, ICLR 2020] Dropedge: towards deep graph convolutional networks on node classification
>
> #### ****Reasons for why sometimes GGCN is not the best****
> First of all, please recall that GGCN is the model that incorporates the 3 strategies into the GCN model. GGCN always performs better than GCN across all the datasets. As for the other models, they use different strategies. Thus from Table 2, we cannot conclude that the designs might be harmful; we can conclude though that, in homophilous settings, some other strategies might be more effective. The good performance of Geom-GCN in homophilous settings can be explained using our findings. Geom-GCN expands the neighborhood by including structurally similar nodes found by other algorithms (such as Isomap, struc2vec). This greatly increases the degree of every node. If every node's degree is increased by the same amount, the nodes with very low relative degrees will benefit. In the homophilous graphs and shallow layers, this graph modification increases the accuracy for low-degree nodes. Thus, in homophilous graphs, we observe that Geom-GCN sometimes performs better than GGCN.
>
> You are right about signed messages not being always helpful. In the discussion under theorem 3, we require the error rate of the sign approximation to be low enough for signed messages to be beneficial. Usually, we compute signs using node representations, so when the quality of representations is low, evidenced by low node classification accuracy, the error of sign approximation is high. In the case when the node classification accuracy is very low, we do not suggest using signed messages.
>
> #### ****Scalability analysis****
> We have added the scalability analysis in ***appendix B.5***. The time complexity of GGCN is similar to attention-based models. Thus, we compare with GAT the total time (training+test) on four larger datasets. In Table B.6, for most cases, GGCN trains faster than GAT because we learn fewer parameters in the degree correction and sign messages. Though the very simple model GCN has smaller time complexity, they perform far worse than GGCN.

---

> > ### Comment · Reviewer_nAQJ · 2021-12-02
> > **Response to rebuttal**
> >
> > Thank you for the response.
> >
> > - For the difference between Table 2 and Table B.1, I would suggest the authors to update the draft to either align these numbers or explicitly describe why they're different.
> > - For the GGCN performance discussion, I also suggest the authors to include the discussion into the draft.

---

> > > ### Author Response · Authors · 2021-12-02
> > > **Response to the feedback**
> > >
> > > Thanks for the suggestions! Though currently, we cannot update the draft anymore as the deadline has passed, we will add the explanations/discussions in the appendix in our next version.

---

### Official Review · Reviewer_osW7 · 2021-11-08

**Correctness:** 2
**Technical Novelty And Significance:** 2
**Empirical Novelty And Significance:** 2
**Recommendation:** 3
**Confidence:** 3

**Main Review:**

Strengths:
 1. The attempt to develop a theory to simultaneously address heterophily and oversmoothing issues in graph neural networks is important.
 2. Evaluation  shows that the proposed GGCN shows some promising results.

Weaknesses and Concerns:

 - Theoretical analysis is based on first on the vanilla linear GCN model, i.e., SCN of (Wu et al 2019), which removes non-linear activations. This makes analysis of GCNs trivial. For example, with the linear GCN model, the statement in Lemma 1 is trivially true.

  - In addition, the graph generation process is also greatly simplified, assuming each node generates "links" uniform-randomly to nodes within the same class vs. nodes of the different class, independently of the choices made by, say, neighboring nodes.

 - Despite these simplifying assumptions (which are Okay to make the problems trackable in order to obtain rigorous theoretical results), I find that the theoretical developments are rather sloppy, and I have difficulty in following the theoretical derivations and proofs. All in all, the paper lacks rigor as I would have liked to see in a "theory"-oriented paper.


  For example, in the proof of Lemma 1, you go from eq.(13) to eq(14), but never explain what $\mathbf{x}$ is? Is $\mathbf{x}$ simply $\mathbf{f}_{i}^{L}$?
Or does it refer to the input (node features)?  I assume it is the latter, as in Lemma 2, you seem to use $\mathbf{x}$ to denote the input. In any case, if it is the latter,  you have NOT established it yet (without accumulating all the weight matrices $\\mathbf{w}$ in the lower layers.

 - The authors use the term "movements" in the latent space rather loosely, I have no ideas what it means precisely. In Lemma 2, it seems that you meant by the shifting the decision boundary.

 As an example, take the proof of Lemma 2. First of all, by stating that the representation of class 1 becomes: $\{\mathbf{f}^{(L)}_{i} - t {\mathbf{w}}^{*T}\}$, you seem to assume that only one hyperplane has shifted, as t ${\mathbf{w}}^{*T}$ represents only a single hyperplane. This is in general NOT true.  Why do you need to"overload"  $f_1(x)$ and $f_2(x)$? This seems to create confusion. In fact, I am not sure what the first equation in eq(15) represents, in particular, what $w'$ and $b'$ represent, since they are never explained. What is the relation between eq.(15) and eq.(16)?

  I am not sure wherer eq.(17) comes from.

 Likewise in the proofs of Theorem 1 and Theorem 2, you start taking expectations without even properly defining the random variables and distributions over which you are taking expectations. Specifically, in the proof of Theorem 2,  when you start considering the higher layers $l$, you take the expectations based only on $k_i$ -- here you completely ignore the fact that once you move to higher layers where the aggregation of  "messages" from neighboring nodes would  introduce dependence on the inputs of the neighboring nodes.

- There are a lot of issues similar to what I have pointed out here. It really makes following the proofs difficult. I have NO confidence in the theoretical results obtained in the paper.

 - In terms of the GGCN model, having "negative" message passing to deal with heterophily seems rather straightforward -- ideally, if the model is designed properly,  this "negativity should follow directly from the model (if one assumes  "+" captures similarity, and "-" captures "dissimilar", as in signed networks); I suspect that the (relative) "low" degree claim is mostly due to the simplistic graph generation process used in the paper where node links are generated independently of other nodes.



**Summary Of The Paper:**

The paper studies two problems plaguing graph neural networks (GCNs/GNNs): 1) oversmoothing of GCNs/GNNs; and 2) GCNs/GNNs often do not yield good performance on heterophilous networks. The authors aim to develop a theory to explain these issues simultaneously, where a few lemmas and theorems were developed based on certain simplifying assumptions. Based on these theoretical results, the authors develop a generalized GCN (GGCN) with negative message passing. Evaluation is conducted and compared with the state-of-the-art.

**Summary Of The Review:**

The papers attempts to develop a theory to simultaneously address heterophily and oversmoothing issues in graph neural networks, which is a plus. However, despite making many simplifying assumptions, the "theoretical" developments are sloppy and hard to follow. As such, I have NOT confidence on the theoretical results obtained in the paper. I have NO confidence in the theoretical results obtained in the paper.

---

> ### Author Response · Authors · 2021-11-22
> **Response to Reviewer osW7 (Part-1)**
>
> Thanks for reading our theories carefully. Based on your suggestions, we have modified the paper to make it more clear and rigorous. We have changed the following parts:
> 1. We change our assumptions so that the theories do not depend on the specific generation processes of graphs and we also slightly change the proofs in the appendix to accommodate for the change. But the conclusions and the final formula used in the main paper do not change. We provide important changes in blue.
> 2. We use $\mathbb{E}_{V1|V2}(\cdot|\cdot)$ to represent that under conditions described by V2, the expectation is taken over V1. We have annotate all the expectations in main text and proofs to make it clear.
> 3. We have added necessary explanations in the proofs to explain how we do it.
> 4. We have modified the proof of theorem 2 in the appendix and we take into account the dependence of representations in higher layers.
>
> ***We kindly remind the reviewer to also take a look at the "relation between theorems and conclusions" part in the "general response to all reviewers" as we have provided the information you are interested in and we think it can help clear some of your questions.***
>
> ### Annotations and confusions:
>
> 1.1 what is $\mathbf{x}$
>
> $\mathbf{x}$ is the vector variable to describe the plane. Any vectors that satisfy the equation are on this plane. $\{{\mathbf{f}\_{i}}^{L}\}$
> are $|V|$ random vectors associated with the nodes, which are different from $\mathbf{x}$. $\mathbf{x}$ and $\{\mathbf{f}_{i}^{L}\}$ are vectors in the same space and the hyperplane is the decision boundary with respect to the input to the last logistic regression layer. We have added this explanation in the appendix.
>
> 1.2 without accumulating all the weight matrices in the lower layers
>
> We mentioned at the beginning of section 3 that we are considering SGC model and in section 2, SGC model does not have weight matrices in lower layers. They use only one weight matrix, which is equivalent to the products of all weight matrices in a linear GCN. We change the wording to make it more clear. For the weight matrix in the last layer, it relates to the decision boundary we are analyzing in Lemmas. We further emphasize it in section 2 of the paper.
>
> 1.3 Movements of node representations
>
> We view the changes of node representations across different layers as continuous movements of vectors. For example, we view node $i$'s representation $\mathbf{f}_i^{(l+1)}$ at layer $l$ as being moved from $\mathbf{f}_i^{(l)}$. We do not refer to the optimal hyperplane, though the optimal hyperplane will change (not limited to shifting) as a response to the change of node representations or movements of node representations. We have changed the wording in the paper.
>
> 1.4 Single hyperplane
>
> Because we are considering binary classification and for binary classification, there is one hyperplane. Besides, for SGC, only one matrix is used and it is used in the last regression layer. Thus, we only have one hyperplane.
>
> 1.5 What does overloading $f_1(x)$ mean?
> When we analyze SGC of different layers, the distributions of the representations, which is the input to the last logistic regression layer, are different. We were overloading the same function because the way we analyze them is the same. Since it creates confusion, we have modified that part in the appendix.
>
> 1.6 Explanation of Lemma 2.
>
> We have clarified some of your confusion in the general response (relation between theorems and conclusions).
> We introduce $\mathbf{w}'$ and $\mathbf{b}'$ in the paragraph before Lemma 2 (highlighted in blue). They represent the parameters which describe the new optimal hyperplane. The new optimal hyperplane changes as a response to the movements(change) of node representations. We add an explanation of Eq.15 before it in the appendix. Eq.15 represents the misclassification rate of nodes in class 1 (probability of nodes in class 1 being misclassified) and Eq.16 represents the misclassification rate of nodes in class 2 (probability of nodes in class 2 being misclassified). Eq.17 represents the total misclassification rate and we have rewritten it in a more detailed and clear way in the appendix.
>
> 1.7 What expectation is taken over
>
> We have added the annotations in both the main text and the appendix to specify what the expectation is taken over.

---

> > ### Author Response · Authors · 2021-12-02
> > **Regarding our response**
> >
> > Dear reviewer:
> >
> > We have made great efforts in updating the theories so that they are more rigorous in the writing. And we also relax our assumptions and modify the proofs accordingly to make the theories more general.
> >
> > We would like to know how you think about the current theory or if there's anything you would like us to clarify or if there's any part you still feel sloppy?

---

> ### Author Response · Authors · 2021-11-22
> **Response to Reviewer osW7 (Part-2)**
>
> ### Theory part
> #### **Assumptions about the graph generation**
> Thanks for your suggestion. In the updated version of our paper, we have modified our assumptions and it does not assume any process of graph generation. Besides, we do not assume the independence of the links now.
>
> #### **Dependence of node representations in higher layers**
> Thanks for the suggestion. We have modified the proof of theorem 2 in the appendix and we take into account the dependence of representations in higher layers based on the modified assumptions.
>
> #### **Linear analysis**
> We agree that we make the linear assumption to make this problem tractable. However, as a start point to explore the relation between the heterophily and oversmoothing problem, the simplified theoretical models do provide us meaningful insights and useful practices. Those insights and practices have empirically shown to be effective in real scenarios: model with non-linearity and multi-class setting. Though theories under more general settings are preferable, the non-linear nature of neural networks makes the full analysis prohibitively hard. Thus, in this work, we adopt the linear assumption as in [Zhu, NeurIPS 2020; Oono, ICLR 2019; Li, AAAI 2018] and the binary classification assumption as in [Baranwa, ICML 2021], and show the applicability of our theoretical results to more general settings in our empirical analysis. Furthermore, most existing analysis for oversmoothing adopts the spectral perspective (they neglect non-linearity and weights), which makes it hard to understand how the performance gradually degenerates and how different kinds of nodes (with different homophily levels and degrees) behave. Our work is a complement to those works.
>
> #### **Signed messages straight-forward and relative degrees may be dependent on how graphs are generated**
> In our modified assumption, we do not assume a specific graph generation process, but we still have the same formula and results, which show that relative degrees are an important factor. Moreover, our degree correction is directly derived from the relative degree and it also shows effectiveness in the experiments. In appendix C3, we show that relative degrees become large through the correction mechanism after learning.
> As for signed messages, it might be easy to understand its benefits to heterophily data, but its benefits to oversmoothing problem are not very straightforward and no one has mentioned it. Also, we show the interplay between the error rate of sign approximation, homophily level, and the relative degree in theorem 3, which is not revealed before. What we point out there, signed messages are not appropriate in cases when the error rate is not high enough. As people normally approximate signs using node representations, we suggest not using signs when the node classification accuracy is not very high.

---

### Public Comment · ~Susheel_Suresh1 · 2021-11-13
**Related work on analyzing GNNs and mixing patterns**

Hi,
I wanted to bring to authors notice a related work from KDD '21 [1] which is missed in the paper and discussions. The referenced paper also analyses the behavior of GNNs w.r.t mixing patterns in networks using the notion of local assortativity. While the analysis is experimental in nature I still think it is very relevant here. Moreover, some of the conclusions obtained by the authors of the ICLR submission have also been witnessed in [1].

Wanted your thoughts on the experimental observations witnessed in [1] in relation to the current submission.


[1] Susheel Suresh, Vinith Budde, Jennifer Neville, Pan Li, and Jianzhu Ma. Breaking the Limit of Graph Neural Networks by Improving the Assortativity of Graphs with Local Mixing Patterns. KDD '21

Thanks

---

### Author Response · Authors · 2021-11-22
**General response to all reviewers (Part-1)**

We thank the reviewers for their thoughtful and constructive feedback. We are glad that reviewer 1EPP finds the problem we tackle enlightening and reviewer ehjS thinks our approach is interesting. In this response, we would like to emphasize the novelty and contribution of this work, and clarify the relation between the theorems and the conclusions. We address the questions raised by each reviewer separately.

## *****Novelty and contribution*****
[All reviewers, esp. reviewer nAQJ, reviewer ehjS] The goal for this work is to reveal the connection between the heterophily and oversmoothing problems, and not to propose another GNN which can alleviate these two problems.******As the first work to investigate the relation (others may study the two problems but they view them independently), we hope that it will encourage future research that considers the two problems together, and designs new powerful GNNs by learning from practices of both sides.******

As a starting point, our theorems point out that both problems can be explained through three cases in Fig. 2 and that in each problem (at different stages) there will be different dominant cases.

As for the architectural designs, though [Tang et al., CIKM 2020] uses degree information to help increase node classification accuracy in homophilous graphs, it's unclear if their design can tackle the heterophily and oversmoothing problems. Also, we use completely different designs as our designs are directly derived from our theories.

For the signed messages, though [Bo et al., AAAI 2021] and [Chien et al., ICLR 2021] use signed aggregation of graph filters, they fail to show that signed messages may help alleviate oversmoothing problem and signed messages may sometimes be harmful. On the other hand, we justify the usage of signed messages based on our framework and we also point out in the discussion under Theorem 3 that signed messages are not always helpful because the error rate of sign approximation also plays an important role. This is not discussed in other papers. Moreover, our theories are not bound to specific design choices. Instead, we point out two possible directions: correcting the low-degree nodes and passing signed messages. In fact, there can be different and more complex design choices for these two directions; we chose to apply simple designs for the sake of verifying the findings and conclusions of our theorems.

---

### Author Response · Authors · 2021-11-22
**General response to all reviewers (Part-2)**

## *****Relation between theorems and conclusions*****
[All reviewers, esp. reviewer osW7, reviewer 1EPP] Thanks for your careful reading of our theorems and proofs. We would like to first address some confusion about the relation between the theorems and conclusions before addressing your detailed questions.

We view the changes of node representations across different layers as continuous movements of vectors. For example, we view node $i$'s representation $\mathbf{f}_i^{(l+1)}$ at layer $l$ as being moved from $\mathbf{f}_i^{(l)}$. Since we analyze the SGC model, they use only one weight matrix (and use it in the last regression layer), which is equivalent to the products of all weight matrices in a linear GCN. The input to the last regression layer decides the misclassification rate of SGC model. Thus we study the quality of the representations sent to the last regression layer and study their decision boundary.

Lemma 1 states that in the multi-class setting, the decision boundaries are a set of hyperplanes. Thus, in the binary classification problem that we studied, the decision boundary is a hyperplane. Lemma 2 addresses how the total misclassification rate will differ if we send in node representations of the $(L+1)$-th layer ($\{\mathbf{f}_i^{(L+1)}\}$) compared to if we send in representations of the $L$-th layer ($\{\mathbf{f}_i^{(L)}\}$). These essentially map to the misclassification rate of an $(L+1)$-layer SGC and the misclassification rate of an $L$-layer SGC.

Since we view $\{\mathbf{f}_i^{(L+1)}\}$ as being moved from $\{\mathbf{f}_i^{(L)}\}$, in Lemma 2 we discuss what will happen to the misclassification rate if we move $\{\mathbf{f}_i^{(L)}\}$ towards/closer to the original optimal decision boundary. We argue that under the "non-swapping" condition, moving towards the original optimal decision boundary will lead to non-decreasing misclassification rate. That is to say, under the "non-swapping" condition, if---after graph convolution---the node representations move towards/closer to the original decision boundary, it has a negative impact on the misclassification rate.

We note that in Lemma 2 we only require $t>0$ (a non-zero step closer to the original decision boundary) and we do not require the expectation of the node representations to move across the origin. In later theorems, we show the factors that will make the representations move closer to the original decision boundary because Lemma 2 already shows moving towards the original decision boundary is undesired.

The theorems convey one main idea: nodes in the graph can be categorized into 3 cases (Fig.2): (i) low homophily, (ii) high homophily, and (iii) high relative degree and high homophily. Only case (iii) is beneficial to the node classification task. And we explain the connection of the two problems through these cases. For different problems and at different stages, the dominant case is different, so they exhibit different behaviors.

We compute the conditional expectation of the representations in the next layer, given the degree of node $v_i$ (any node we are interested in) and some information about representations in the current layer. Through this way, we can see how nodes with

---

### Author Response · Authors · 2021-11-22
**General response to all reviewers (Part-3): Summary of changes**

We sincerely appreciate the constructive suggestions that the reviewers have provided us. We have modified the version of our paper and we summarize the modifications as follows (Important changes have been marked blue in the text):

****Theories****:

1. We relax our assumptions so that the theories do not depend on the specific generation processes of graphs and we also change the proofs in the appendix to accommodate for the relaxed assumptions. But the conclusions and the final formula used in the main paper do not change. We provide important changes in blue.

2. We use
$\mathbb{E}_{V1|V2}(\cdot|\cdot)$
to represent that under conditions described by V2, the expectation is taken over V1. We have annotated all the expectations in main text and proofs to make it clear.

3. We have added necessary explanations in the proofs to explain how we do it.

4. We have modified the proof of theorem 2 in the appendix and we take into account the dependence of representations in higher layers.

****Experiments****:

1.  We have added a section (appendix B.4) to use the three cases (Fig.2) to analyze the datasets and we find that our characterization is more informative to predict the performance of GCN than using global heterophily for the graphs

2. We have added scalability analysis (appendix B.5).

****Writtings****

1. We have added the citations suggested by the reviewer

2. We have changed the wording based on the feedback from the reviewers

---

### Author Response · Authors · 2021-11-29
**A kind reminder to review our responses**

Dear reviewers:

As the final discussion stage will end soon (Nov. 29), would you please take a look at our responses and let us know if you have further concerns? Could you leave comments and your feedback about our responses, and let us know if our responses address your questions or if you still have doubts about anything?

We appreciate your help!

---

### Decision · Program_Chairs · 2022-01-20

**Decision:**

Reject

**Comment:**

This paper focuses on investigating the relations between the heterophily and over-smoothness problem. However, the relationship is not clear.

The over-smoothness problem considers the features and the adjacency matrix, while the heterophily incorporates the adjacency matrix and the labels. They have different views on the graph. It may not be treated as the same coin. Besides, the stacked aggregations lead to indistinguishable node representations and poor performance in the over-smoothing problem. The same phenomenon appears in the heterophily problem because the features in different classes are falsely mixed, leading to indistinguishable nodes [2]. They have the same phenomenon but different origins. It may be not a necessity to combine these two problems.

Besides, MADGap[1] is proposed to evaluate the over-smoothness problem. It is unreliable to use the accuracy and the degree to measure this problem. Therefore, in section 3, the relations between node degrees and the homophily ratio cannot infer the relations between the heterophily and over-smoothness problem.

As a result, the authors should carefully re-organize their paper and results.

A suggestion is to pack the submission as a new method to learn from heterophily instead of trying to make such a close relationship with over-smoothing.

- [1] Measuring and Relieving the Over-smoothing Problem for Graph Neural Networks from the Topological View. AAAI 2020
- [2] Beyond homophily in graph neural networks: Current limitations and effective designs. NeurIPS 2020